# Biphasic regulation of osteoblast development via the ERK MAPK–mTOR pathway

**Jung-Min Kim[1], Yeon-Suk Yang[1], Jaehyoung Hong[2], Sachin Chaugule[1], Hyonho Chun[2], Marjolein CH van der Meulen[3,4], Ren Xu[5,6], Matthew B Greenblatt[4,7], Jae-hyuck Shim[1,8,9]***

[1]Department of Medicine, University of Massachusetts Medical School, Worcester, United States; [2]Department of Mathematical Sciences, Korea Advanced Institute of Science and Technology, Daejeon, Republic of Korea; [3]Meinig School of Biomedical Engineering and Sibley School of Mechanical & Aerospace Engineering, Cornell University, Ithaca, United States; [4]Research Division, Hospital for Special Surgery, New York, United States; [5]State Key Laboratory of Cellular Stress Biology, School of Medicine, Xiamen University, Fujian, China; [6]Fujian Provincial Key Laboratory of Organ and Tissue Regeneration, School of Medicine, Xiamen University, Xiamen, Fujian, China; [7]Department of Pathology and Laboratory Medicine, Weill Cornell, New York, United States; [8]Horae Gene Therapy Center, Worcester, United States; [9]Li Weibo Institute for Rare Diseases Research, University of Massachusetts Chan Medical School, Worcester, Worcester, United States

**\*For correspondence:**
jaehyuck.shim@umassmed.edu

**Abstract** Emerging evidence supports that osteogenic differentiation of skeletal progenitors is a key determinant of overall bone formation and bone mass. Despite extensive studies showing the function of mitogen-activated protein kinases (MAPKs) in osteoblast differentiation, none of these studies show in vivo evidence of a role for MAPKs in osteoblast maturation subsequent to lineage commitment. Here, we describe how the extracellular signal-regulated kinase (ERK) pathway in osteoblasts controls bone formation by suppressing the mechanistic target of rapamycin (mTOR) pathway. We also show that, while ERK inhibition blocks the differentiation of osteogenic precursors when initiated at an early stage, ERK inhibition surprisingly promotes the later stages of osteoblast differentiation. Accordingly, inhibition of the ERK pathway using a small compound inhibitor or conditional deletion of the MAP2Ks *Map2k1* (MEK1) and *Map2k2* (MEK2), in mature osteoblasts and osteocytes, markedly increased bone formation due to augmented osteoblast differentiation. Mice with inducible deletion of the ERK pathway in mature osteoblasts also displayed similar phenotypes, demonstrating that this phenotype reflects continuous postnatal inhibition of late-stage osteoblast maturation. Mechanistically, ERK inhibition increases mitochondrial function and SGK1 phosphorylation via mTOR2 activation, which leads to osteoblast differentiation and production of angiogenic and osteogenic factors to promote bone formation. This phenotype was partially reversed by inhibiting mTOR. Our study uncovers a surprising dichotomy of ERK pathway functions in osteoblasts, whereby ERK activation promotes the early differentiation of osteoblast precursors, but inhibits the subsequent differentiation of committed osteoblasts via mTOR-mediated regulation of mitochondrial function and SGK1.

## Editor's evaluation

This work provides a novel insight into regulation of osteogenesis by ERK-mTOR pathway. The authors proposed that the effect of Erk pathway would be mediated mTOR2-SGK1. The mitochondrial metabolisms appears to be involved in this regulation. This study is well performed, and the manuscript is clearly written.

## Introduction

Signal transduction and transcription programs are coordinated to determine the commitment, proliferation, and differentiation of skeletal progenitors to osteoblasts. Once skeletal progenitors are committed to osteoprogenitors, they become preosteoblasts, proliferate, and differentiate into mature osteoblasts producing mineral and extracellular matrix proteins. Mature osteoblasts lastly terminally differentiate into osteocytes and embed themselves within the bone matrix (*Salhotra et al., 2020*). Proliferation is dominant at an early stage of osteogenesis, while proliferation rates decrease and extracellular mineralization increases during osteoblast differentiation (*Infante and Rodríguez, 2018*; *Long, 2011*). The balance between proliferation and differentiation during osteogenesis is tightly regulated by the sequential activation of signaling cascades, such as the mitogen-activated protein kinase (MAPK) and the mammalian/mechanistic target of rapamycin (mTOR) pathways.

The extracellular signal-regulated kinase (ERK) MAPK pathway transduces activation signals from extracellular cues, such as bone morphogenic proteins (BMPs) and wingless-related integration site (WNT) ligands, determining the specification of skeletal progenitors to osteoprogenitors (*Ge et al., 2007*; *Greenblatt et al., 2013*; *Kim et al., 2019*; *Matsushita et al., 2009*). Extensive studies have demonstrated the importance of the ERK MAPK pathway in promoting early commitment and differentiation of skeletal progenitors to the osteoblast lineage and skeletal mineralization (*Greenblatt et al., 2022*). For example, conditional deletion of *Mapk1* (ERK2) and *Mapk2* (ERK1) in the mesenchyme (*Prrx1^Cre*) causes severe limb deformity and bone defects due to impaired osteogenesis (*Matsushita et al., 2009*). Similarly, mice lacking the MAP2Ks in the ERK pathway, *Map2k1* (MAP2K1, MEK1) and *Map2k2* (MAP2K2, MEK2), in osteoprogenitors (*Sp7^Cre*) showed low bone mass and a cleidocranial dysplasia-like phenotype (CCD), similar to that seen in mice and humans with runt-related transcription factor 2 (*RUNX2*) haploinsufficiency (*Ge et al., 2007*; *Ge et al., 2009*; *Kim et al., 2019*). Thus, ERK-mediated activation of RUNX2 is required for the commitment of skeletal progenitors to osteoprogenitors and the subsequent proliferation of osteoprogenitors at early differentiation stages. Additionally, the ERK MAPK pathway mediates phosphorylation of ribosomal s6 kinase 2 (RSK2; *Yang et al., 2004*). *Rsk2*-deficient mice and humans are osteopenic due to impaired production of type I collagen, the major organic component of bone. *RSK2* mutations have been reported to be associated with Coffin–Lowry syndrome showing prominent skeletal dysplasia features (*David et al., 2005*; *Hanauer and Young, 2002*). As neurofibromin 1 (NF1) functions as a negative regulator of the ERK MAPK pathway, loss of function mutations in *NF1* cause neurofibromatosis type 1 (NF1) syndrome, characterized by a complex set of bone phenotypes that can include osteopenia and impaired fracture healing as well as cutaneous neurofibromas (*Bok et al., 2020*; *Crawford and Schorry, 1999*; *de la Croix Ndong et al., 2014*). Similar to NF1, gain of function mutations in protein-tyrosine phosphatase nonreceptor-type 11 (PTPN11, SHP2) enhance ERK signaling in osteoblasts, causing Noonan syndrome associated with impaired skeletal mineralization and short stature (*Binder, 2009*; *Choudhry et al., 2012*; *Roberts et al., 2007*). It remains unclear how modulation of ERK signaling can produce these diverse and clinically important phenotypes. These considerations motivate consideration of how the role of ERK may vary based on context. In particular, the role of ERK in regulating the stages of osteoblast differentiation and function after lineage commitment remains unresolved.

In addition to RUNX2, the ERK MAPK pathway controls the mTOR pathway, a signaling modulator of mitochondrial biogenesis that regulates cellular energy metabolism (*Morita et al., 2013*). Cross-regulation between these pathways in various cancer cells has been also reported (*Mendoza et al., 2011*). ERK inhibition by an MEK inhibitor activates the mTORC2–AKT signaling axis downstream of epidermal growth factor (EGF; *Yu et al., 2002*), and a similar increase in AKT activation by ERK inhibition was reported in anaplastic lymphoma kinase-addicted neuroblastoma (*Umapathy et al., 2017*). Mechanistically, impaired ERK activation by the MEK inhibitor trametinib treatment enhanced phosphorylation of the mTORC2 complex subunits, Rictor or stress-activated protein kinase (SAPK)-interacting protein 1 (SIN1), via activation of induced integrin-linked kinase, resulting in AKT activation

(*Pairo-Castineira et al., 2021*; *Umapathy et al., 2017*). Thus, determining the function and the regulatory mechanism of ERK in osteoblast differentiation and bone formation is clinically significant as the ERK pathway is the most frequently mutated pathway in cancer and, accordingly, ERK pathway components, including rat sarcoma (RAS), rapidly accelerated fibrosarcoma, and MEK isoforms, are all well-established targets to treat a wide variety of ERK pathway-activated cancers (*Hong et al., 2020*; *Hyman et al., 2015*; *Robert et al., 2015*; *Tiacci et al., 2021*). Given that many of these patients are at an increased risk for fracture, it is important to weigh the risks and benefits when determining the role of the ERK pathway in the regulation of bone mass and the potential skeletal impact of ERK pathway-directed therapies.

Here, we establish dynamic roles for the ERK MAPK pathway during osteogenesis, with the ERK–mTOR signaling axis acting in osteoblasts to regulate mitochondria-dependent energy metabolism and SGK1-mediated production of pro-angiogenic and osteogenic factors. ERK inhibition in mature osteoblasts promotes bone formation via mTOR-mediated activation of mitochondria and SGK1. Accordingly, inhibition of the mTOR pathway using rapamycin attenuated bone accrual in MEK1/2-deficient mice. Our study uncovers a surprising dichotomy of ERK pathway functions in osteoblasts, with the ERK pathway promoting bone formation in osteoblast precursors but inhibiting bone formation in committed osteoblasts via mTOR pathway.

## Results

### A biphasic role for the ERK MAPK pathway in osteogenesis

To investigate the role of the ERK MPAK pathway in osteoporosis, the MEK inhibitor trametinib was treated to mice with estrogen deficiency-induced bone loss. Twelve-week-old mice were subjected to the ovariectomy (OVX) model of postmenopausal osteoporosis or sham surgery and were subsequently treated with trametinib via oral gavage for 8 weeks. While OVX surgery induced a significant bone loss in the femur of vehicle-treated mice, femoral bone loss was prevented by treatment with trametinib, as shown by a greater trabecular bone volume (Tb. BV) and trabecular thickness (Tb. Th; *Figure 1A*). Remarkably, dynamic histomorphometry analysis demonstrated an increase in bone formation rate per bone surface (BFR/BS) and osteoblast surface on the trabecular bone area (Ob.S/BS) of trametinib-treated OVX mice (*Figure 1B, C*). Thus, ERK inhibition prevents estrogen deficiency-induced bone loss by promoting bone formation due to augmented osteoblast activity.

To gain insights into the mechanism underlying this finding, we examined the effects of ERK inhibition on the proliferation and differentiation of osteoblast precursors using human bone marrow-derived mesenchymal stromal cells (BMSCs). Expression of key osteogenic commitment factors, *RUNX2* and *SP7* (Osterix), was markedly upregulated 8 days after osteogenic culture as expected, establishing that the BMSCs at this time point represent committed osteoblast-lineage cells (*Figure 1—figure supplement 1*). ERK inhibition starting at day 8 of culture promoted osteoblast differentiation, whereas ERK inhibition from day 0 of culture conversely inhibited proliferation and osteogenic potential of BMSCs (*Figure 1D, E*). Similar findings were observed with committed osteoblasts isolated from mouse calvarium, which also showed increased osteogenic differentiation in the presence of trametinib. This was evident in ERK inhibitor induced increases in alkaline phosphatase (ALP), mineralization, and expression of osteogenic genes, including integrin-binding sialoprotein (*Ibsp*) and osteocalcin (*Bglap; Figure 1F–H*). Notably, cell proliferation rates were largely unaffected (*Figure 1I*). These results suggest that while the ERK MAPK pathway plays a positive role in the early stages of osteogenesis, it acts as a negative regulator of the later stages of osteoblast differentiation. Thus, ERK inhibition in late-stage osteoblasts by trametinib treatment is likely to be responsible for the increased osteoblast activity observed in ERK inhibitor-treated OVX mice.

### Impaired ERK signaling in mature osteoblasts promotes bone formation

To confirm these findings using genetic approaches in vivo, *Map2k1* (MEK1) and *Map2k2* (MEK2), which are upstream of ERK1/2, were conditionally deleted in mature osteoblasts and osteocytes by crossing with *Dmp1$^{Cre}$* (Dentin matrix protein1) mice, which targets conditional gene deletion to mature osteoblasts and osteocytes (*Lu et al., 2007*). Deletion of MEK1/2 was confirmed in RNA from the tibiae of WT and *Map2k1$^{fl/fl}$;Dmp1$^{Cre}$;Map2k2$^{-/-}$* (double knockout, dKO$^{Dmp1}$) mice (*Figure 2A*),

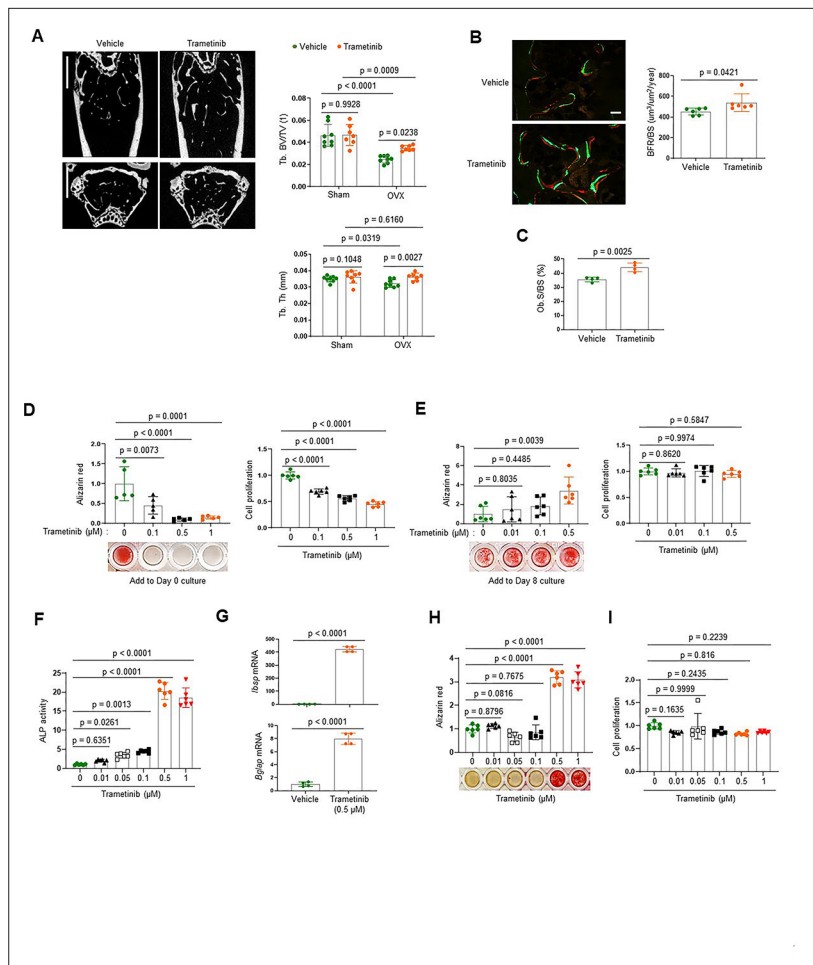

**Figure 1.** The extracellular signal-regulated kinase (ERK) mitogen-activated protein kinase (MAPK) pathway plays biphasic roles in osteogenesis. (**A–C**) Twenty-week-old female mice were treated with vehicle or trametinib after either Sham or ovariectomy (OVX) surgery. Representative microCT images of the femoral bone (A, left) and quantification of relative trabecular bone volume/total volume (Tb. BV) and thickness (Tb. Th, A, right) are displayed. Alternatively, dynamic histomorphometry analysis of the femoral bones was performed to assess in vivo osteoblast activity in OVX group. Representative images of calcein/alizarin labeling (B, left) and quantification of bone formation rate (BFR)/bone surface (BS, B, right) and osteoblast surface (Ob.S)/bone surface BS, (**C**) are displayed. Scale bars, 1 mm (**A**), 50 µm (**B**). Human bone marrow-derived mesenchymal stromal cells (BMSCs) were treated with different doses of trametinib at days 0 (**D**) or 8 (**E**) of osteogenic culture. Mineralization and cell proliferation were assessed by alizarin red and alamar blue staining after 14 days of culture, respectively. (**F–I**) Mouse wildtype osteoblasts (Obs) were treated with different doses of trametinib at day 0 of osteogenic culture and 6 days later, alkaline phosphatase (ALP) activity (**F**) and osteogenic gene expression by real-time PCR (**G**) were assessed. Mineralization by alizarin red staining (**H**) and cell proliferation (**I**) by alamar blue staining were assessed after 18 days of culture. Data are representative of three independent experiments (A) [left], (B) [left], (**D–I**) or pooled from two experiments (A) [right], (B) [right], (**C**). A two-tailed unpaired Student's *t*-test for comparing two groups (**A–C, G**) or ordinary one-way analysis of variance (ANOVA) with Dunnett's multiple comparisons test (**D–F, H, I**) (A–I; error bars, standard deviation [SD] of biological replicates).

The online version of this article includes the following source data and figure supplement(s) for figure 1:

**Source data 1.** Source data for *Figure 1A–I*.

**Figure supplement 1.** Osteogenic gene expression in human bone marrow-derived mesenchymal stromal cells (BMSCs).

**Figure supplement 1—source data 1.** Source data for *Figure 1—figure supplement 1*.

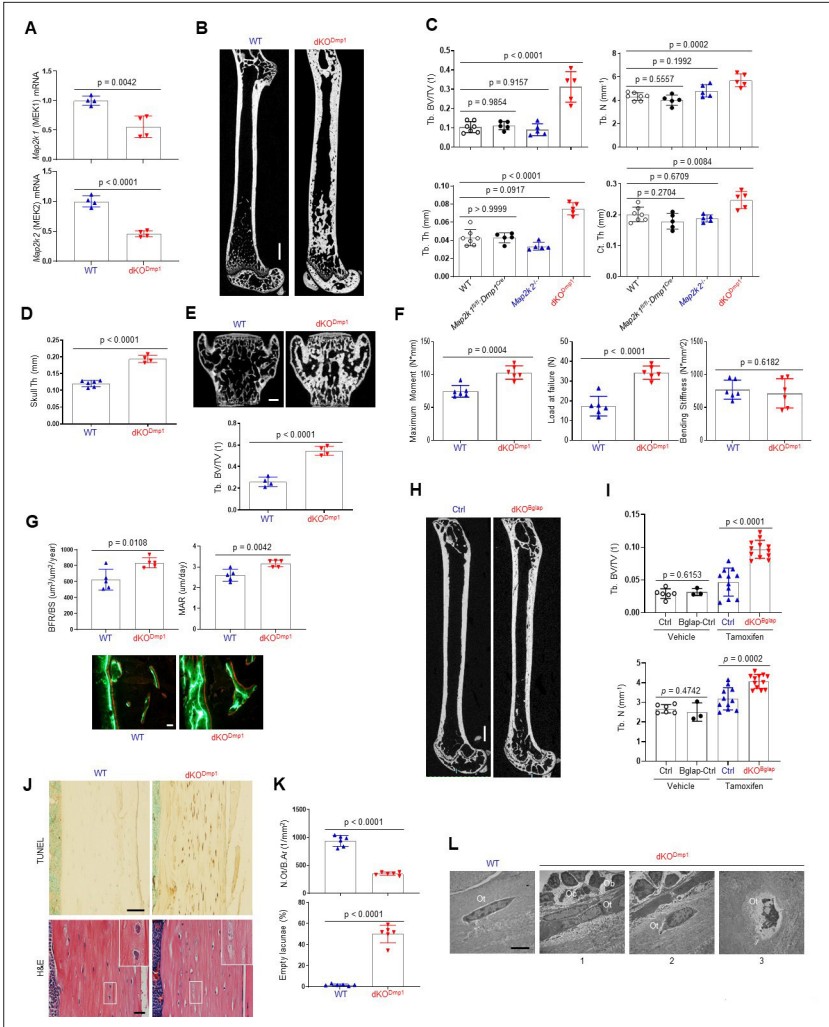

**Figure 2.** MEK1/2 deletion in mature osteoblasts promotes bone formation in mice. (**A**) *Map2k1* (MEK1) and *Map2k2* (MEK2) mRNA levels in the tibial bones of *Map2k1^fl/fl^;Map2k2^+/+^* (WT) and *Map2k1^fl/fl^;Dmp1^Cre^;Map2k2^-/-^* (dKO^Dmp1^) mice were measured by RT-PCR analysis. (**B, C**) MicroCT analysis showing femoral bone mass in 8-week-old *Map2k1^fl/fl^;Map2k2^+/+^* (WT), *Map2k1^fl/fl^;Dmp1^Cre^;Map2k2^+/+^* (*Map2k1^fl/fl^;Dmp1^Cre^*), *Map2k1^fl/fl^;Map2k2^-/-^* (*Map2k2^-/-^*), and *Map2k1^fl/fl^;Dmp1^Cre^;Map2k2^-/-^* (dKO^Dmp1^) mice. 3D reconstruction (**B**) and the relative quantification (**C**) are displayed. Trabecular bone volume/total volume (Tb. BV/TV), trabecular thickness (Tb. Th), trabecular number Tb. (N), and cortical thickness (Ct.Th). Scale bar, 1 mm (**B**). MicroCT analysis of 8-week-old WT and dKO^Dmp1^ skull (**D**) and vertebrae (**E**). Quantification of skull thickness Th (**D**), and trabecular bone mass (**E**, bottom) and 3D reconstruction of lumbar 4 (L4) vertebrae (**E**, top) are shown. Scale bar, 200 μm (**E**). (**F**) Biomechanical properties of 8-week-old WT and dKO^Dmp1^ femurs, including maximum moment, load at failure, and bending stiffness were quantified using three-point bending test. (**G**) Dynamic histomorphometry analysis of 8-week-old WT and dKO^Dmp1^ femurs. Quantification (top) and images of calcein/alizarin labeling (bottom) are displayed. BFR/BS, bone formation rate per bone surface; MAR, mineral apposition rate. Scale bar, 50 μm (bottom). (**H, I**) MicroCT analysis of femoral bone mass in 17-week-old *Map2k1^fl/fl^;Map2k2^-/-^* and *Map2k1^fl/fl^;Bglap^CreErt^;Map2k2^-/-^* mice treated with vehicle or tamoxifen; vehicle-treated *Map2k1^fl/fl^;Map2k2^-/-^* (Ctrl) and *Map2k1^fl/fl^;Bglap^CreErt^;Map2k2^-/-^* (Bglap-Ctrl) mice, tamoxifen-treated *Map2k1^fl/fl^;Map2k2^-/-^* (Ctrl) and *Map2k1^fl/fl^;Bglap^CreErt^;Map2k2^-/-^* (dKO^Bglap^) mice. 3D reconstruction (**H**) and the relative quantification (**I**) are displayed. Scale bar, 1 mm (**H**). (**J**) TUNEL (top) and hematoxylin and eosin (H&E) (bottom)-stained longitudinal sections of 8-week-old WT and dKO^Dmp1^ femurs. Scale bars, 50 μm (top) and 20 μm (bottom). (**K**) Numbers of osteocytes/bone area (N.Ot/B.Ar) and empty lacunae in the cortical bones of 8-week-old WT and dKO^Dmp1^ femurs. (**L**) Transmission electron microscopy (TEM) images of osteocytes in the cortical bones of 8-week-old WT and dKO^Dmp1^ femurs. 1, Ot in osteoids; 2, Ot in mineralized bone matrix; 3, apoptotic Ot in bone matrix. Ot, osteocyte; Oc, osteoclast; Ob, osteoblast. Scale bar, 5 μm. Data are representative of three independent experiments (**A, B, E**) [top], (**G**) [bottom], (**H, J, L**) or pooled from two

*Figure 2 continued on next page*

*Figure 2 continued*

experiments C, D, E [bottom], F, G [top], (**I, K**). A two-tailed unpaired Student's *t*-test for comparing two groups (**A, D–G, I, K**) or ordinary one-way analysis of variance (ANOVA) with Dunnett's multiple comparisons test (**C**) (**A, C–G, I, K**; error bars represent the standard deviation [SD of biological replicates]).

The online version of this article includes the following source data and figure supplement(s) for figure 2:

**Source data 1.** Source data for *Figure 2A, C–G1, K*.

**Figure supplement 1.** Analysis of skeletal phenotypes in dKO$^{Dmp1}$ mice.

**Figure supplement 1—source data 1.** Source data for *Figure 2—figure supplement 1A–C*.

and resulted in a significant increase in femoral bone mass, as evidenced by greater trabecular bone volume (Tb. BV), thickness (Tb. Th), number (Tb. N), and cortical thickness (Ct. Th; *Figure 2B, C*). Likewise, skull thickness and L4 vertebrae trabecular bone volume/tissue volume (Tb. BV/TV) were markedly increased (*Figure 2D, E*). These skeletal phenotypes require the deletion of both MEK1 and MEK2 alleles as deletion of MEK1 (*Map2k1$^{fl/fl}$;Dmp1$^{Cre}$*) or MEK2 (*Map2k2$^{-/-}$*) alone did not show an increase in bone mass compared to WT femurs (*Figure 2C*), suggesting a redundant role for MEK1 and MEK2 in osteoblasts. Consistent with the increased bone mass observed, a three-point bending test of dKO$^{Dmp1}$ mice revealed an increase in maximum bending moment and load at failure, demonstrating enhanced bone mechanical strength (*Figure 2F*). Thus, the bone formed by ERK pathway ablation in late-stage osteoblasts has the biomechanical properties of mature, physiologic bone and not that of fracture-prone pathologic bone. Finally, dynamic histomorphometry analysis demonstrated a significant increase in BFR/BS and mineral apposition rate (MAR) of dKO$^{Dmp1}$ femurs, indicating enhanced in vivo osteoblast activity in the absence of MEK1/2 (*Figure 2G*). Therefore, deletion of MEK1/2 in late-stage osteoblast-lineage cells augmented their bone-forming activity and bone accrual and strength. Importantly, in vivo osteoclast activity and numbers were markedly elevated in dKO$^{Dmp1}$ mice, as shown by an increase in tartrate-resistant acid phosphatase (TRAP)-positive osteoclasts, bone erosion surface, and serum levels of the bone resorption marker C-terminal telopeptide type I collagen (CTx-I; *Figure 2—figure supplement 1A, B*). This was accompanied with reduced expression of osteoprotegerin (OPG, *Tnfrsf11b*) without any alteration in receptor activator of nuclear factor kappa-Β ligand (RANKL, *Tnfsf11*) expression (*Figure 2—figure supplement 1C*), suggesting that the upregulation of RANKL/OPG ratio may be responsible for the enhanced osteoclast activity observed in these mice.

To confirm that deletion of MEK1/2 in mature osteoblasts is responsible for the development of these phenotypes, we generated inducible, osteoblast-specific MEK1/2-knockout mice by crossing *Map2k1$^{fl/fl}$;Map2k2$^{-/-}$* mice with *Bglap$^{CreERT}$* (osteocalcin) mice expressing a tamoxifen-induced Cre recombinase in mature osteoblasts (*Maes et al., 2010*; *Map2k1$^{fl/fl}$; Bglap$^{CreErt}$;Map2k2$^{-/-}$*, dKO$^{Bglap}$). Treatment of mice with tamoxifen (dKO$^{Bglap}$) significantly increased trabecular bone mass of the femur (*Figure 2H,I*), demonstrating that the inhibitory role of the ERK MAPK pathway is specific to mature osteoblasts and reflects the continuous function of ERK in these cells and is not secondary to a developmental abnormality.

Given previous studies showing the importance of the ERK MAPK pathway in osteocyte survival (*Plotkin et al., 2005*; *Ru and Wang, 2020*), terminal deoxynucleotidyl transferase dUTP nick-end labeling (TUNEL) staining was performed in dKO$^{Dmp1}$ femurs to assess osteocyte cell death (*Figure 2J*, top). Osteocyte apoptosis was substantially increased in the cortical bone of dKO$^{Dmp1}$ mice relative to control mice, which corresponded to decreased osteocyte numbers and increased empty lacunae (a cavity within the bone; *Figure 2J*, bottom, K). Likewise, transmission electron microscopy (TEM) analysis of dKO$^{Dmp1}$ femurs demonstrated that terminally differentiated osteocytes embedded in the bone matrix undergo apoptosis (*Figure 2 L3*) while osteocytes at early (*Figure 2 L1*) and intermediate (*Figure 2 L2*) differentiation stages were grossly normal in the absence of MEK1/2 (*Figure 2L*). This is consistent with RT-PCR analysis showing that mRNA levels of early osteocyte markers, including dentin matrix protein1 (*Dmp1*) and phosphate regulating endopeptidase homolog X-linked (*Phex*) were comparable between control and dKO$^{Dmp1}$ tibias. However, expression of the late osteocyte marker, sclerostin (*Sost*), was significantly decreased (*Figure 2—figure supplement 1C*). These results suggest that ERK activation is required for the survival of late/fully mature osteocytes, but not the transition of mature osteoblasts to early osteocytes. Taken together, these results demonstrate that

ERK inhibition in mature osteoblasts promotes bone formation, while impaired ERK signaling in late osteocytes results in cell death.

## ERK inhibition increases osteogenesis and production of angiogenic and osteogenic factors

As seen in dKO$^{Dmp1}$ mice, Cre-induced deletion of MEK1 and MEK2 MAP2Ks in committed osteoblasts increased ALP activity, mineralization, and osteogenic gene expression. Deletion of MEK1 or MEK2 alone did not impact osteogenic differentiation (*Figure 3A–C*, *Figure 3—figure supplement 1A*). While proliferation rates of control osteoblasts were markedly increased under osteogenic conditions, little to no increase was observed in the absence of MEK1/2 (*Figure 3D*), suggesting that ERK inhibition may control a shift toward differentiation at the expense of proliferation. To gain insights into the mechanisms by which ERK inhibition promotes osteoblast differentiation, we performed whole transcriptome analysis of dKO osteoblasts 6 days after cell growth (GM) or osteogenic (OIM) culture conditions. Gene ontology analysis demonstrated that, under cell growth conditions, upregulated genes were highly enriched in the pathways involved in ossification, skeletal system morphogenesis, and connective tissue development (*Figure 3E*), consistent with an increase in ALP activity and expression of osteogenic genes in undifferentiated dKO osteoblasts (*Figure 3—figure supplement 1B–F*). These results suggest that ERK inhibition induces spontaneous osteogenic differentiation by upregulating transcription of genes favorable to osteogenic differentiation. Under osteogenic conditions, gene ontology analysis of differentiated dKO osteoblasts revealed upregulation of genes associated with extracellular matrix organization, while genes related to cell proliferation, including nuclear division and cell cycle, were downregulated (*Figure 3F*, *Figure 3—figure supplement 2A–D*). These results suggest that impaired ERK signaling in later stage committed osteoblasts upregulates transcriptional programs associated with osteogenic differentiation while suppressing the induction of genes associated with cell proliferation. Intriguingly, the genes associated with the pro-osteogenic WNT/β-catenin and TGF-β signaling pathways were highly upregulated in dKO osteoblasts under osteogenic, but not undifferentiated, conditions (*Figure 3—figure supplement 3A–C*). This is consistent with luciferase assays showing a significant increase in activities of key transcription factors of WNT and TGF-β signaling, β-catenin and Smads, in the absence of MEK1/2 under osteogenic, but not undifferentiated, conditions (*Figure 3—figure supplement 4A*). Likewise, protein levels of β-catenin and Smad3 in dKO cells cultured under osteogenic conditions were markedly increased (*Figure 3—figure supplement 4B*), suggesting that ERK inhibition at later stages of osteogenic differentiation enhances the pro-osteogenic WNT and TGF-β signaling pathways. Accordingly, pharmacological inhibition of these signaling pathways reversed enhanced osteogenesis of dKO cells, as shown by reduced ALP activity and osteogenic gene expression in a dose-dependent manner (*Figure 3—figure supplement 4C–F*). These results suggest that enhanced WNT and TGF-β signaling at later stages of osteogenic differentiation of dKO osteoblasts might be major contributors to drive osteogenic differentiation program.

While the expression and transcriptional activity of RUNX2 and SP7 were not strongly impacted in mature dKO osteoblasts (*Figure 3—figure supplement 5A–C*), ERK inhibition in mature osteoblasts markedly increased the expression and transcription activity of ATF4 (OSE1-luc; *Figure 3—figure supplement 5D, E*). These results suggest that during the later stages of osteoblast differentiation, the ERK MAPK pathway is required for post-translational modifications of ATF4, but not RUNX2 and SP7. This finding provides a clear mechanistic distinction between the role of the ERK pathway in early osteogenic commitment, which relates to regulation of RUNX2 and SP7, from the role of the ERK pathway in later stage osteoblasts which, by contrast, includes regulation of signaling pathways related to ATF4, WNT/β-catenin, and TGF-β.

Whole transcriptome analysis of dKO osteoblasts also revealed significant upregulation of secreted cytokines and/or growth factors involved in ossification and angiogenesis/vasculature development (*Figure 3F*), such as fibroblast growth factor 9 (*Fgf9*; **Behr et al., 2010**), ectonucleotide pyrophosphatase/phosphodiesterase 2 (*Enpp2*; **Cholia et al., 2015**), transforming growth factor-beta 2 (*Tgfb2*; **Wu et al., 2016**), insulin-like growth factors 1, 2 (*Igf1, 2*; **Yakar et al., 2018**), bone morphogenic proteins 1, 3 (*Bmp1, 3*; **Wu et al., 2016**), C-C Motif Chemokine Ligand 5 (*Ccl5*; **Suffee et al., 2012**), and *Ccl11* (**Salcedo et al., 2001**; *Figure 3G*). Accordingly, conditioned medium (CM), collected from dKO osteoblasts, markedly increased osteogenic potentials of wildtype BMSCs (*Figure 3H*) and capillary tube formation of bone marrow-derived endothelial progenitor outgrowth cells (EPOCs, *Figure 3I*).

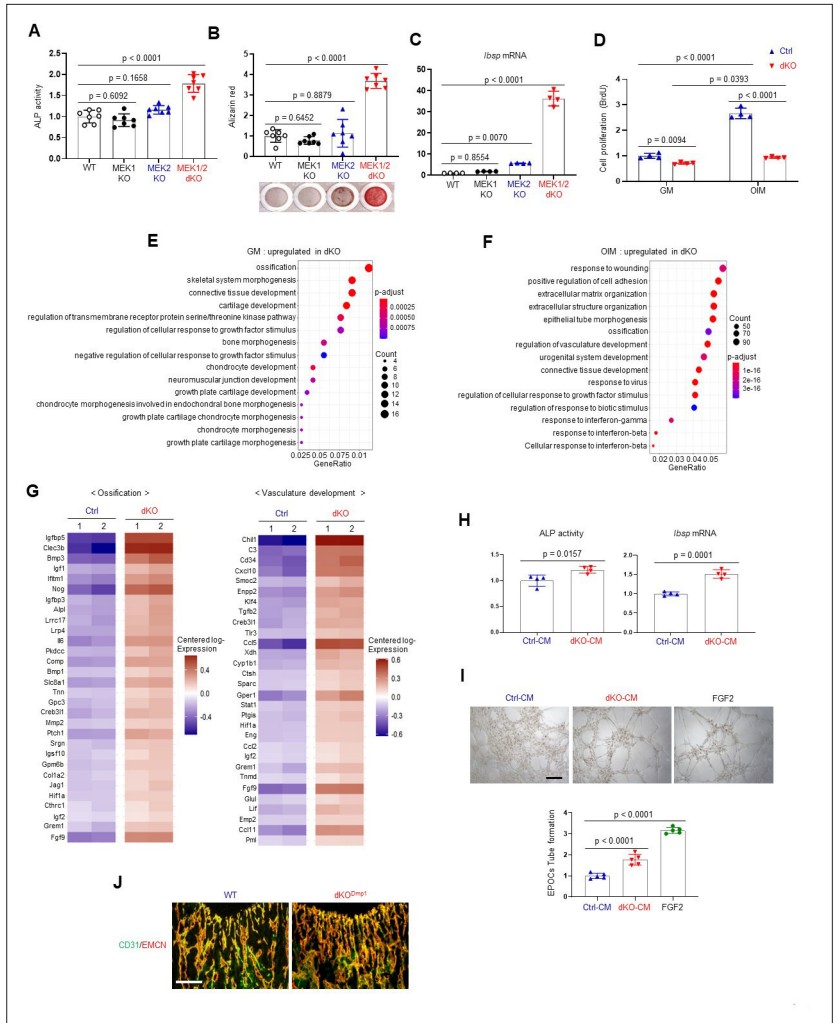

**Figure 3.** Effects of MEK1/2 deletion on osteoblast differentiation and production of angiogenic and osteogenic factors. (**A–C**) Mouse *Map2k1fl/fl;Map2k2+/+* and *Map2k1fl/fl;Map2k2−/−* osteoblasts (Obs) were infected with control vector or Cre recombinase-expressing lentiviruses; *Map2k1fl/fl;Map2k2+/+* Obs with control (WT) or Cre (MEK1 KO), *Map2k1fl/fl;Map2k2−/−* Obs with control (MEK2 KO) or Cre (MEK1/2 dKO). Puromycin-selected Obs were cultured under osteogenic conditions and alkaline phosphatase (ALP) activity (**A**) and osteogenic gene expression (**C**) were determined at day 6 and mineralization (**B**) was analyzed at day 18 of culture. (**D**) MEK2 KO (Ctrl) and MEK1/2 dKO (dKO) Obs were cultured with control growth medium (GM) or osteogenic induction medium (OIM) and cell proliferation was analyzed by Bromodeoxyuridine (BrdU) incorporation at day 6 of the culture. (**E, F**) Transcriptome analysis of Ctrl and dKO Obs 6 days after GM or OIM culture. Biological process output of gene ontology analysis was performed in both GM and OIM group for upregulated genes in dKO relative to Ctrl Obs. The color indicates adjusted p value as estimated by the Benjamini–Hochberg method with the threshold of significance p = 0.05 and q = 0.005. (**G**) Heatmaps for ossification- and vasculature-associated gene expression. The top 30 upregulated genes in dKO Obs relative to Ctrl Obs are displayed as each row and column represent gene symbol and sample, respectively. The log10 expression (read count) was centered across samples and red and purple denote upregulated and downregulated, respectively. (**H**) Conditioned medium (CM) from Ctrl and dKO Obs was collected at day 6 under osteogenic culture and mouse wildtype bone marrow-derived mesenchymal stromal cells (BMSCs) were cultured under osteogenic condition in the presence of CM of Ctrl Obs and dKO Obs and ALP activity (left) and *Ibsp* mRNA level (right) were assessed at day 6. (**I**) Capillary tube formation of mouse endothelial cells (EPOCs) was performed in the presence of CM for 5 hr. Representative images (top) and quantification for the number of branches are displayed (bottom). FGF2 was used as a positive control. Scale bar, 200 µm. (**J**) Immunofluorescence for CD31 (green) and endomucin (EMCN, red) in the epiphyseal area of 8-week-old WT and dKODmp1 femurs. Scale bar, 100 µm. Data are representative of three independent experiments (**A–D, H–J**). For transcriptome analysis, biological duplicates were analyzed (**E–G**). Ordinary one-way analysis of variance (ANOVA)

*Figure 3 continued on next page*

*Figure 3 continued*

with Dunnett's multiple comparisons test (**A–C, I**) or a two-tailed unpaired Student's *t*-test for comparing two groups (**D, H**) (**A–D, H, I**; error bars, standard deviation [SD] of biological replicates).

The online version of this article includes the following source data and figure supplement(s) for figure 3:

**Source data 1.** Source data for *Figure 3A–D, H1*.

**Figure supplement 1.** Characterization of MEK1/2-deficient osteoblast-lineage cells.

**Figure supplement 1—source data 1.** Source data for *Figure 3—figure supplement 1A–F*.

**Figure supplement 2.** Transcriptome analysis of MEK1/2-deficient osteoblast-lineage cells.

**Figure supplement 3.** MEK1/2-deficient osteoblasts show gene enrichment of WNT and TGF-β signaling.

**Figure supplement 4.** MEK1/2-deficient osteoblasts show enhanced WNT/β-catenin and TGF-β signaling at late stages of osteogenic differentiation.

**Figure supplement 4—source data 1.** Source data for *Figure 3—figure supplement 4A, C–F*.

**Figure supplement 4—source data 2.** Full immunoblots for *Figure 3—figure supplement 4B*.

**Figure supplement 5.** Effects of MEK1/2 deletion on RUNX2 expression and transcriptional activity in osteoblasts.

**Figure supplement 5—source data 1.** Full immunoblots for *Figure 3—figure supplement 5A*.

**Figure supplement 5—source data 2.** Source data for *Figure 3—figure supplement 5B–E*.

Likewise, an immunofluorescence analysis of dKO$^{Dmp1}$ femurs showed elevated levels of CD31- and endomucin-positive skeletal vasculature supporting osteoblast development (*Xu et al., 2018*; *Figure 3J*). These results suggest that pro-angiogenic and/or -osteogenic factors secreted from dKO osteoblasts enhance angiogenesis and osteogenesis, promoting bone formation. Taken together, MEK1/2 deficiency does not only increase osteoblast differentiation by upregulating osteogenic transcriptional programs, but also improves bone-forming environment by producing pro-angiogenic and/or -osteogenic factors.

## ERK inhibition enhances mitochondrial energy metabolism in osteoblasts

It has been well established that the ERK MAPK pathway controls cellular metabolism in cancer cells and highly proliferating stem cells (*Lee et al., 2017*; *Papa et al., 2019*). Mature osteoblasts produce collagen for extracellular matrix formation using adenosine triphosphate (ATP) as a major energy resource (*Lee et al., 2017*). Moreover, glucose and glutamine, used for mitochondrial ATP production, are important for cell proliferation at early stages of osteoblast differentiation and matrix mineralization at later differentiation stages (*Karner et al., 2015*; *Wei et al., 2015*). Thus, we hypothesized that ERK-mediated regulation of glucose/glutamine metabolism may be involved in the determination of osteogenic potential and differentiation. Remarkably, the differentiation of dKO osteoblasts was substantially reduced by glutaminase inhibition but only modestly impacted by glucose deprivation (*Figure 4A, B*), demonstrating the importance of glutamine metabolism in ERK-mediated regulation of osteogenesis. Specifically, the responsiveness of dKO osteoblasts to the glutaminase inhibitor (bis-2-[5-phenylacetamido-1,3,4-thiadiazol-2-yl]ethyl sulfide or BPTES) was greater than that of control osteoblasts, as shown by a significant reduction in ALP activity, cell proliferation, and osteogenic gene expression in the presence of BPTES (*Figure 4B, C*). Likewise, these cells showed elevated levels of glutaminase (*Gls*), a key enzyme that produces glutamate from glutamine (*Figure 4D*), and increased glutamate production and release (*Figure 4E*), suggesting enhanced glutamine metabolism in dKO osteoblasts. Of note, in contrast to previous studies showing critical roles of glucose metabolism (*Wei et al., 2015*), effect of glucose depravation on osteoblast maturation was intact in the absence of MEK1/2 (*Figure 4A*), suggesting that the ERK MAPK pathway is not required for glucose metabolism during the later stages of osteoblast differentiation. This finding provides a clear mechanistic distinction between the role of the ERK pathway in early osteogenic commitment which relates to regulation of glucose metabolism, from the role of the ERK pathway in later stage osteoblasts which, by contrast, includes regulation of glutamine metabolism.

Given the importance of glutamine metabolism in mitochondria-mediated energy production, we next examined the intracellular ATP content in dKO osteoblasts and found increased ATP levels (*Figure 4F*). Additionally, mitochondrial numbers and membrane potentials and a ratio of

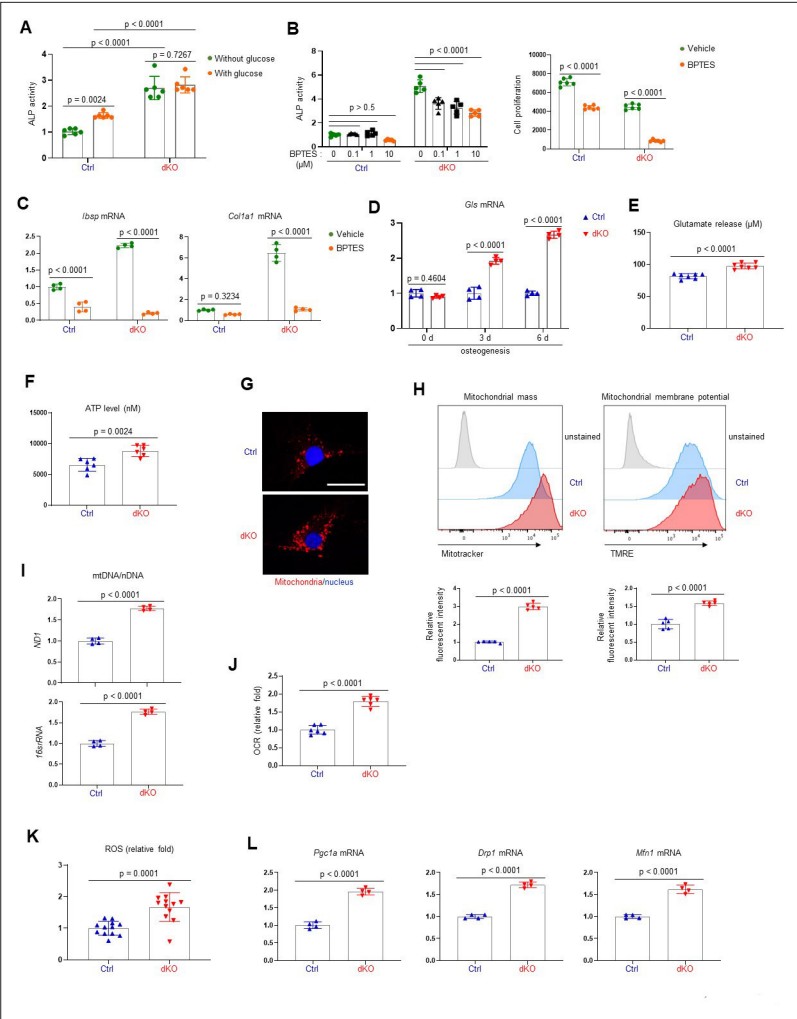

**Figure 4.** MEK1/2 deletion enhances mitochondria-mediated energy metabolism in osteoblasts. (**A**) Ctrl and dKO Obs were cultured under osteogenic conditions in the presence or absence of glucose and 6 days later, alkaline phosphatase (ALP) activity was assessed. Ctrl and dKO Obs were treated with different doses of bis-2-[5-phenylacetamido-1,3,4-thiadiazol-2-yl]ethyl sulfide (BPTES) under osteogenic conditions and 6 days later, ALP activity and cell proliferation (**B**) and osteogenic gene expression (**C**) were assessed. mRNA levels of *Gls* in Ctrl and dKO Obs were measured by RT-PCR analysis at different time points of osteogenic culture (**D**) and levels of glutamate in the supernatant were measured for extracellular glutamate release at day 6 culture of osteogenesis (**E**). (**F–I**) Intracellular adenosine triphosphate (ATP) levels (**F**), mitochondrial numbers (**G**), and the ratio of mitochondrial DNA (mtDNA, mt-*ND1* or mt-*16sRNA*) to nuclear DNA (nDNA, Hk2) (**I**) were assessed in Ctrl and dKO Obs after 6 days of culture. Alternatively, flow cytometry was used to measure mitochondrial numbers and membrane potential of Ctrl and dKO Obs treated with Mitotracker or TMRE at day 4 of osteogenic culture, respectively (**H**). Numbers indicate median fluorescence intensity (H, bottom). Scale bar, 75 µm (**G**). (**J**) Oxygen consumption rate (OCR) in Ctrl and dKO Obs 6 days after osteogenic culture. (**K**) Intracellular reactive oxidative species (ROS) levels in Ctrl and dKO Obs were assessed 6 days after osteogenic culture. (**L**) mRNA levels of mitochondria-related genes in Ctrl and dKO Obs were assessed by RT-PCR analysis. Data are representative of three independent experiments. A two-tailed unpaired Student's *t*-test for comparing two groups (**A, B**) [right], (**C–F, H–L**) or ordinary one-way analysis of variance (ANOVA) with Dunnett's multiple comparisons test (**B** [left]) (**A–F, H–L**; error bars, standard deviaiton [SD] of biological replicates).

The online version of this article includes the following source data and figure supplement(s) for figure 4:

**Source data 1.** Source data for *Figure 4A–F, H–L*.

**Figure supplement 1.** Extracellular signal-regulated kinase (ERK) inhibition enhances mitochondrial function in osteoblasts.

**Figure supplement 1—source data 1.** Source data for *Figure 4—figure supplement 1A*.

mitochondrial DNA (mtDNA) levels to nuclear DNA (nDNA) levels in dKO or trametinib-treated osteo-blasts were all markedly increased compared to control or vehicle-treated osteoblasts (*Figure 4G–I*, *Figure 4—figure supplement 1*). This corresponds to enhanced oxygen consumption rate (OCR), an indicator of cellular respiration, in dKO osteoblasts (*Figure 4J*), demonstrating that ERK inhibition promotes mitochondrial function and metabolic activity in committed osteoblasts. Finally, levels of reactive oxidative species (ROS; *Figure 4K*) and mRNA levels of peroxisome proliferator-activated receptor gamma coactivator 1-alpha (*Pgc1a*), dynamin-related protein 1 (*Drp1*), and mitofusin 1 (*Mfn1; Figure 4L*), genes that are important for mitochondrial biogenesis, fission, and fusion, were also markedly elevated in dKO osteoblasts. Taken together, these results suggest that ERK inhibition increases osteoblast differentiation via augmented glutamine metabolism and mitochondrial function.

## ERK inhibition promotes bone formation via the mTOR pathway in osteoblasts

As a metabolic hub in response to nutrients and growth factors, the mTOR pathway plays a crucial role in mitochondria-mediated cell metabolism during skeletal development and homeostasis (*Chen and Long, 2018*). Two main mTOR signaling complexes, mTORC1 and mTORC2, are composed of discrete protein-binding partners to regulate growth, motility, and metabolism in osteoblasts. mTORC1 activation mainly promotes osteoblast proliferation and differentiation in response to bone anabolic signals and their regulatory processes (*Chen et al., 2014*; *Lim et al., 2016*). On the other hand, mTORC2 activation not only mediates osteogenesis and skeletal mineralization in response to growth factors or mechanical loading, but it also plays a role in the restoration of metabolic homeostasis in response to nutrient fluctuations (*Szwed et al., 2021*). Additionally, mTORC2 has been reported to be associated with age-related osteoporosis since impaired mTORC2 activation in aged mice switches the fate of skel-etal progenitors to adipocytes from osteoblasts (*Lai et al., 2016*). Given that ERK functions upstream of the mTOR pathway in regulating mitochondria-mediated energy metabolism (*Morita et al., 2013*), we tested the ability of mTOR inhibition to reverse the enhanced osteogenic differentiation seen after MEK1/2 deletion (*Figure 5A*). Rapamycin was used to inhibit both mTORC1 and mTORC2 activation (*Lamming et al., 2012*) in dKO osteoblasts, demonstrating that rapamycin treatment attenuated ALP activity (*Figure 5B*) and expression of osteogenic genes (*Figure 5C*) in the absence of MEK1/2. Of note, in contrast to many contexts where rapamycin solely inhibits the mTORC1 pathway, including p70S6 kinase (p70S6K), rapamycin treatment was effective at inhibiting the phosphorylation of the mTORC2 downstream molecules, protein kinase B (AKT) and serum/glucocorticoid regulated kinase 1 (SGK1), in dKO osteoblasts (*Figure 5—figure supplement 1*). These results suggest that rapamycin treatment decreases the differentiation of dKO osteoblasts by inhibiting both mTORC1 and mTOCR2 pathways. To test this hypothesis in vivo, 2-week-old dKO[Dmp1] mice were treated with rapamycin for 6 weeks and femoral bone mass was assessed in 8-week-old mice using microCT. While little effect of rapamycin on femoral bone mass was seen in control mice, trabecular bone mass and cortical bone thickness were both decreased in dKO[Dmp1] mice when treated with rapamycin relative to vehicle, as shown by a significant reduction in trabecular bone volume (Tb. BV), number (Tb. N), thickness (Tb. Th), and cortical thickness (Ct.Th) (*Figure 5D, E*). These results suggest, as observed in dKO osteoblasts, mTOR inhibition can reverse the enhanced bone-forming activity by MEK1/2-deletion in mice. A histo-logic analysis of rapamycin-treated dKO[Dmp1] femurs revealed that mTOR inhibition was also effective in preventing the osteocyte apoptosis occurring in the absence of MEK1/2 (*Figure 5F*). Additionally, osteoclast surface (Oc.S) in the trabecular bone was slightly decreased in these femurs, suggesting that rapamycin treatment may directly suppress osteoclast differentiation and activity (*Glantschnig et al., 2003*; *Kneissel et al., 2004*; *Figure 5G*). Taken together, impaired ERK signaling is likely to promote osteogenic differentiation and bone formation via activation of the mTOR pathway, whose inhibition can reverse dKO[Dmp1] phenotypes in vitro and in vivo.

## ERK inhibition promotes osteoblast differentiation via mTORC2/SGK1 activation

To understand how ERK controls mTOR activation in osteoblasts, we examined the effects of MEK1/2 deficiency on mTORC1 and mTORC2 signaling. While mTORC1 primarily promotes cell growth via activation of p70S6K and inactivation of eIF4E-binding protein 1 (4EBP1), mTORC2 regulates cell survival and proliferation via AKT phosphorylation as well as cellular metabolism via activation of

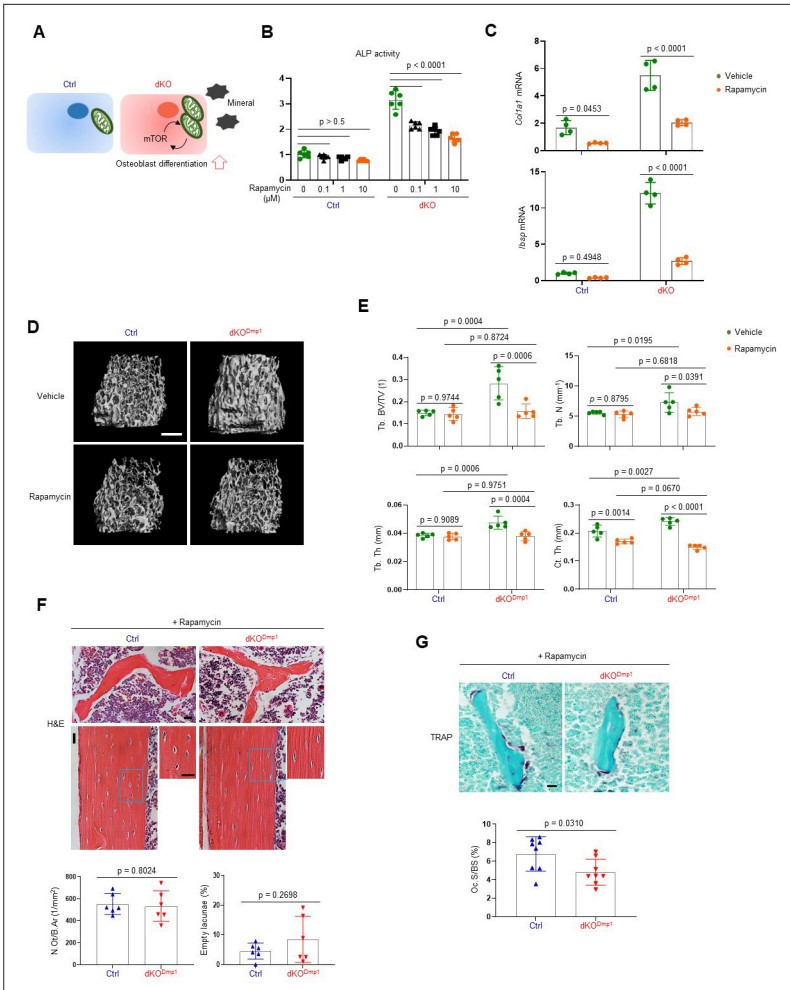

**Figure 5.** Rapamycin treatment reverses skeletal phenotypes of MEK1/2-deficient mice. (**A**) Diagram showing the mechanisms by which extracellular signal-regulated kinase (ERK) inhibition enhances osteoblast differentiation due to augmented mechanistic target of rapamycin (mTOR)-mediated mitochondrial function. Ctrl and dKO Obs were treated with different doses of rapamycin under osteogenic conditions and 6 days later, alkaline phosphatase (ALP) activity (**B**) and osteogenic gene expression (**C**) were determined. (**D, E**) MicroCT analysis showing femoral bone mass in 8-week-old Ctrl and dKO$^{Dmp1}$ mice treated with vehicle or rapamycin. 3D reconstruction (**D**) and the relative quantification (**E**) are displayed. Scale bar, 500 µm (**D**). (**F**) Hematoxylin and eosin (H&E)-stained longitudinal sections of 8-week-old Ctrl and dKO$^{Dmp1}$ femurs treated with rapamycin. Representative images (top) and numbers of osteocytes/bone area (N.Ot/B.Ar) and empty lacunae (bottom) are displayed. Scale bar, 20 µm (top). (**G**) Tartrate-resistant acid phosphatase (TRAP)-stained longitudinal sections of 8-week-old Ctrl and dKO$^{Dmp1}$ femurs treated with rapamycin. Representative images (top) and osteoclast surface/bone surface (Oc.S/BS) (bottom) are displayed. Scale bar, 20 µm (top). Data are representative of two or three independent experiments (B–D, F [top], G [top]) or pooled from two experiments (E, F [bottom], G [bottom]). Ordinary one-way analysis of variance (ANOVA) with Dunnett's multiple comparisons test (**B**) or a two-tailed unpaired Student's *t*-test for comparing two groups (**C, E–G; B, C, E–G**; error bars, standard deviation [SD] of biological replicates).

The online version of this article includes the following source data and figure supplement(s) for figure 5:

**Source data 1.** Source data for *Figure 5B, C, E–G*.

**Figure supplement 1.** Rapamycin inhibits both mTORC1 and mTORC2 pathways.

**Figure supplement 1—source data 1.** Full immunoblots for *Figure 5—figure supplement 1*.

SGK1 and N-Myc downstream regulated 1 (NDRG1) (*Saxton and Sabatini, 2017*). In comparison to control osteoblasts, dKO osteoblasts showed a significant increase in phosphorylation levels of mTORC2 downstream molecules, including AKT, SGK1, and NDRG1, whereas little to no alteration in the phosphorylation levels of the mTORC1 downstream molecules p70S6K and 4EBP1 were detected

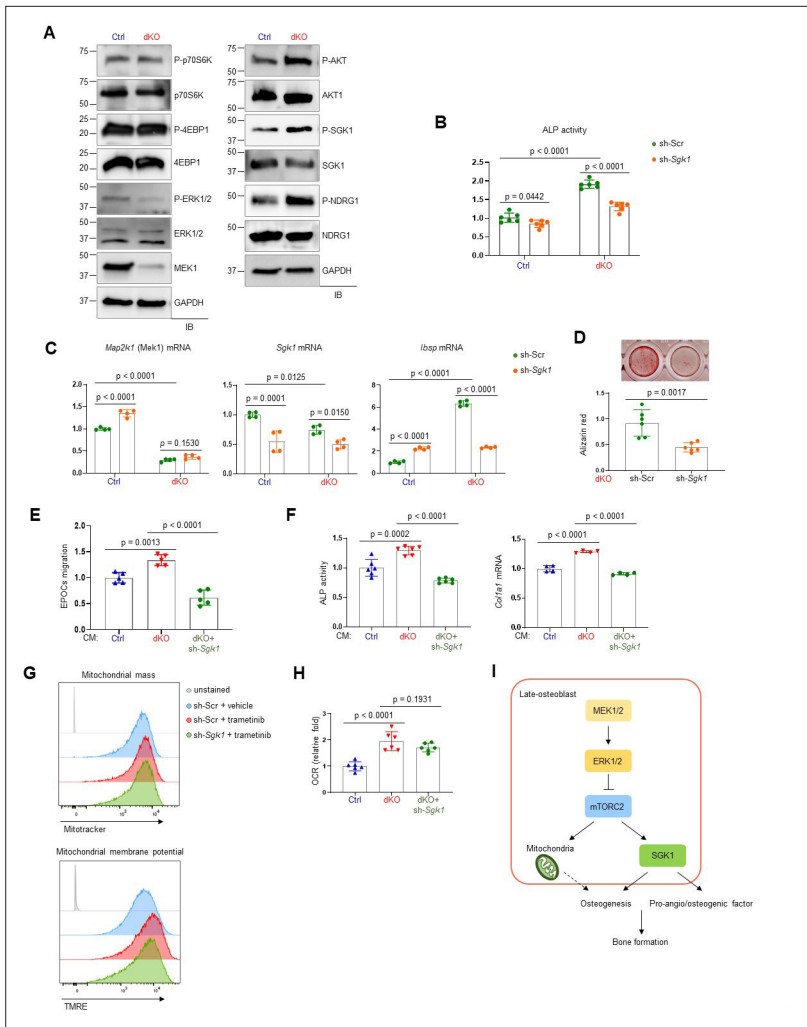

**Figure 6.** mTORC2/SGK1 activation is important for MEK1/2 deficiency-induced osteoblast differentiation.
(**A**) Immunoblot analysis showing phosphorylation levels of mechanistic target of rapamycin (mTOR) signaling components in Ctrl and dKO Obs. GAPDH was used as a loading control. (**B–D**) Ctrl and dKO Obs expressing sh-Scr or sh-*Sgk1* shRNAs were cultured under osteogenic conditions. Alkaline phosphatase (ALP) activity (**B**), osteogenic markers expression (**C**), and mineralization (**D**) were assessed after 6 or 18 days of culture, respectively. (**E**) Conditioned medium (CM) collected from Ctrl and dKO Obs expressing sh-Scr or sh-*Sgk1* shRNAs 6 days after osteogenic culture was added to transwell migration of mouse endothelial cells (EPOCs) and migrated cells were assessed 12 hr after incubation. (**F**) Mouse bone marrow-derived mesenchymal stromal cells (BMSCs) were cultured under osteogenic condition in the presence of CM, and ALP activity and *Cola1* mRNA level were analyzed at day 6 of the culture. (**G**) Vehicle- or trametinib-treated Obs expressing sh-Scr or sh-*Sgk1* shRNAs were treated with Mitotracker or TMRE 4 days after osteogenic culture and mitochondrial mass and membrane potential were assessed using flow cytometry, respectively. (**H**) Oxygen consumption rate (OCR) in Ctrl and dKO Obs expressing sh-Scr or sh-*Sgk1* shRNAs after 6 days of culture. (**I**) Diagram showing the molecular actions of extracellular signal-regulated kinase (ERK) on bone formation. Data are representative of two or three independent experiments (**A–H**). A two-tailed unpaired Student's *t*-test for comparing two groups (**B–D**) or ordinary one-way analysis of variance (ANOVA) with Dunnett's multiple comparisons test (**E, F, H; B–F, H**; error bars, standard deviation [SD] of biological replicates).

The online version of this article includes the following source data for figure 6:

**Source data 1.** Full immunoblots for *Figure 6A*.

**Source data 2.** Source data for *Figure 6B–F, H*.

in the absence of MEK1/2 (*Figure 6A*). Together with previous studies (*Brown et al., 2020*; *Umapathy et al., 2017*), these results suggest that ERK inhibition in osteoblasts activates mTORC2- but not mTORC1-mediated signal transduction via phosphorylation of mTORC2 subunits, Rictor or SIN1.

In a previous study, SGK1, that functions downstream of mTORC2, played an important role in osteogenic trans-differentiation and calcification of vascular muscle cells (*Voelkl et al., 2018*). Since the genes associated with the ossification pathway in dKO osteoblasts were highly upregulated (*Figure 3E, F*), we hypothesized that ERK inhibition may upregulate osteoblast differentiation by activating the mTORC2–SGK1 signaling axis. Similar to rapamycin-mediated mTOR inhibition (*Figure 5B, C*), *Sgk1* knockdown markedly attenuated ALP activity (*Figure 6B*), expression of osteogenic genes (*Figure 6C*), and mineralization (*Figure 6D*) in dKO osteoblasts, not in control osteoblasts, suggesting that SGK1 functions downstream of ERK during osteoblast differentiation. Notably, the enhanced expression and transcription activity of ATF4 in mature dKO osteoblasts were reversed by *Sgk1* knockdown or mitochondrial inhibitor treatment, suggesting that mTOR–SGK1 and mitochondrial function control ERK-mediated regulation of ATF4 in mature osteoblasts (*Figure 3—figure supplement 4D, E*). Additionally, the ability of CM collected from dKO osteoblasts to promote angiogenesis of EPOCs and osteogenesis of wildtype BMSCs was ablated by *Sgk1* knockdown, as shown by a significant decrease in migration of EPOCs (*Figure 6E*), and ALP activity and *Col1a1* mRNA levels of BMSCs (*Figure 6F*). Notably, little to no alteration of mitochondrial numbers and membrane potentials (*Figure 6G*) and OCR (*Figure 6H*) was observed in trametinib-treated *Sgk1*-deficient osteoblasts or *Sgk1*-deficient dKO osteoblasts, suggesting that SGK1 is dispensable for ERK-mediated mitochondrial function. Taken together, the ERK–mTORC2 signaling axis mediates osteogenesis and production of pro-angio/osteogenic factors via SGK1 activation, while controlling osteogenesis via mitochondria-dependent energy metabolism (*Figure 6I*).

## Discussion

This study demonstrates essential roles for the ERK–mTOR pathway in osteoblast development and function. While ERK activation is crucial for cell proliferation and osteogenic potential at the early stages of osteogenic differentiation, ERK-mediated regulation of the mTOR pathway determines osteogenic differentiation in later stages. In addition, ERK activation dampens the secretion of pro-angiogenic and osteogenic factors required to create an optimal microenvironment for bone formation. Mechanistically, impaired ERK signaling in committed osteoblasts enhances osteoblast differentiation via an increase in mitochondria-mediated metabolic demands, activation of mTORC2–SGK1 signaling axis, thereby promoting bone formation (*Figure 6I*). This corresponds to previous reports showing that mTORC2 activation is required for mitochondrial function and homeostasis (*Wang et al., 2021*) and that both glycolysis and oxidative phosphorylation by mitochondria are required to meet the energy demands needed for bone formation by osteoblasts (*Lee et al., 2017*). However, despite previous studies showing the importance of SGK1 in osteogenic trans-differentiation of vascular smooth muscle cells (*Voelkl et al., 2018*) and energy metabolism in different cellular types and functions (*Mason et al., 2021*), our findings suggest that while ERK–mTORC2–SGK1 signaling axis is required for osteogenesis and production of pro-angiogenic and -osteogenic factors, it is dispensable for mitochondrial function. Thus, our study identified two distinct pathways downstream of the ERK–mTORC2 signaling axis controlling bone formation, mitochondria- and SGK1-mediated osteogenesis and production of pro-angiogenic and -osteogenic factors. Notably, while we demonstrated the mTORC2–SGK1 signaling axis as a critical downstream pathway for ERK during late-stage differentiation, the ERK MAPK pathway is likely to have additional substrates that contribute to its regulation of osteogenesis. Since WNT ligands can activate both the ERK and mTOR pathways independently to regulate skeletal development and homeostasis (*Chen and Long, 2018*; *Karner and Long, 2017*; *Shim et al., 2013*), the ERK and mTORC2–SGK1 signaling axis may also function in parallel at different stages of osteogenesis. Additionally, further investigation will be necessary to define the mechanism by which the ERK MAPK pathway suppresses mTORC2 activation in committed osteoblasts via phosphorylation of mTORC2 subunits, Rictor or SIN1 (*Brown et al., 2020*; *Umapathy et al., 2017*).

ERK activation is dynamically changed in response to various extracellular stimuli during tissue development, homeostasis, and regeneration (*Lavoie et al., 2020*; *Patel and Shvartsman, 2018*). The ERK MAPK pathway plays a critical role in the regulation of cell quiescence/proliferation, differentiation, and commitment of adult stem cells and progenitors, as ERK activation promotes cell

proliferation while suppressing cell differentiation (*Krishnan et al., 2021*). However, very few in vivo studies have reported dynamics of ERK signaling in the skeletal system. This study demonstrates distinct roles of the ERK MAPK pathway at different stages of osteogenesis. In particular, during late-stage osteogenic differentiation, ERK activation is important for mitochondria-mediated glutamine metabolism rather than glucose metabolism (*Figure 4*). Previous studies have shown that osteogenic precursors not only consume glutamine to generate ATP but also synthesize metabolites to promote osteoblast viability and matrix mineralization (*Yu et al., 2019*). Our findings reveal that impaired ERK signaling enhances mitochondria-mediated glutamine metabolism, thereby promoting osteogenic differentiation. Remarkably, this can be reversed by the mTOR inhibitor rapamycin in vitro and in vivo. Since systemic administration of rapamycin increases bone anabolic activity in both normal bone mass and osteopenic conditions (*Bateman et al., 2019*; *Chagin, 2016*; *Izawa et al., 2017*), rapamycin-induced attenuation of osteogenic differentiation and bone anabolic activity in MEK1/2-deficient mice and osteoblasts is unlikely to result from nonspecific or general effects of rapamycin unrelated to the specific defects in MEK1/2-deficient mice. Together with previous studies showing the association of the mTOR pathway with energy metabolism during skeletal development and homeostasis (*Chen and Long, 2018*), our data suggest that the ERK MAPK pathway controls energy metabolism in committed osteoblasts via the mTOR pathway.

This study also justifies an examination of the clinical impact of ERK pathway inhibitors on bone mass and fracture risk, as the biphasic complex contribution of the ERK pathway to bone formation makes empiric study of this question important. This is especially true given that the ERK pathway is the most frequently mutated pathway in cancer (*Guo et al., 2020*), and thus ERK inhibitors are important tools for targeted therapy in a wide range of tumor types. This study also shows that dual treatment with ERK and mTOR inhibitors may be associated with a strong decrease in bone mass, as mTOR inhibition would ablate the anabolic effect of ERK inhibition in late-stage osteoblasts, leaving only the negative effects of ERK on early osteogenic differentiation and commitment. In mice, trametinib treatment increased bone mass in osteoporotic bone (*Figure 1A–C*), indicating that the pro-anabolic effect of ERK inhibition in mature osteoblasts is dominant over the anti-anabolic effects on early-stage osteoblasts. However, the relative dominance of these opposing early versus late-stage effects may depend highly on dose and duration of therapy, in addition to clinical context and patient demographics. Thus, empirical study of the clinical effects of trametinib and other ERK pathway inhibitors will be important, especially given that many cancer patients being treated with these agents will be at increased risk for skeletal fractures due to the effects of either cancer, chemotherapy, or patient demographics impacting bone mass. Given the emerging role of osteoblasts in promoting and maintaining skeletal metastases, our finding that the ERK pathway has complex effects on early- and late-stage osteoblast-lineage cells also suggests that further examination of the impact of ERK pathway inhibition of skeletal metastases will be important to understand the risks and benefits of ERK pathway inhibition (*Bado et al., 2021*; *Swami et al., 2017*; *Zhang et al., 2021*).

## Materials and methods
### Cell culture, antibodies, and reagents
Human BMSCs were purchased from Cyagen Biosciences and were maintained in GM (HUXMX-90011) and cultured in osteogenic medium (GUXMX-90021). Primary osteoblasts were isolated from the calvaria of 5-day-old mouse pups using collagenase type II (50 mg/ml, Worthington, LS004176)/dispase II (100 mg/ml, Roche, 10165859001). MEK1/2-deficient osteoblasts were obtained from calvaria of 5-day-old *Map2k1*<sup>fl/fl</sup>;*Map2k2*<sup>−/−</sup> neonates, transduced with lentiviruses expressing vector control or Cre recombinase, and selected with puromycin. Osteoblasts were maintained in α-minimal essential medium (Gibco) containing 10% fetal bovine serum (Corning), 2 mM L-glutamine (Corning), 1% penicillin/streptomycin (Corning), and 1% nonessential amino acids (Corning) and differentiated with ascorbic acid (200 µM, Sigma, A8960) and β-glycerophosphate (10 mM, Sigma, G9422).

Antibodies specific to MEK1 (12671), P-ERK1/2 (4376), P-p70S6K (9205), P-4EBP1 (2855), P-AKT (4060), P-NDRG1 (5482), β-catenin (8480), and RUNX2 (12556) were purchased from Cell Signaling. Antibodies specific to P-SGK1 (44–1264 G) and GAPDH (CB1001) were purchased from Thermo Scientific and EMD Millipore, respectively. Antibodies specific to ERK1/2 (A0228), p70S6K (A2190), 4EBP (A1248), AKT (A11016), NDRG1 (A2142), SGK1 (A1025), and SMAD3 (A199115) were purchased

from ABclonal. Trametinib (S2673) and BPTES (5301) were purchased from Selleck Chemicals and Tocris Bioscience, respectively. Rapamycin (sc-3054) for cell culture and antimycin A (sc-20246) were purchased from Santa Cruz Biotechnology. WNT inhibitor (IWR-1-endo, 72562) and TGF-β inhibitor (SB431542, 72232) were purchased from STEMCELL Technologies. All constructs encoding shRNAs were purchased from Sigma.

## Mice

*Map2k1*<sup>fl/fl</sup> mice and *Map2k2*<sup>–/–</sup> mice were generated as previously reported, respectively (*Bélanger et al., 2003*; *Bissonauth et al., 2006*) and maintained on a 129/SvEv background. To generate mature osteoblast/osteocyte-specific double knockout (*Map2k1*<sup>fl/fl</sup>;*Dmp1*<sup>Cre</sup>;*Map2k2*<sup>–/–</sup>, dKO<sup>Dmp1</sup>), *Map2k1*<sup>fl/fl</sup>;*Map2k2*<sup>–/–</sup> mice were crossed with *Dmp1*<sup>Cre</sup> mice (*Lu et al., 2007*). Sex- and age-matched littermates were used as controls for all skeletal analyses. To generate mice that harbored inducible deletion of *Map2k1* (MEK1) in mature osteoblasts, *Map2k1*<sup>fl/fl</sup>;*Map2k2*<sup>–/–</sup> mice were crossed with *Bglap*<sup>CreERT</sup> (osteocalcin) mice (*Park et al., 2012*) (*Map2k1*<sup>fl/fl</sup>;*Bglap*<sup>CreERT</sup>;*Map2k2*<sup>–/–</sup>, dKO<sup>Bglap</sup>). For postnatal activation of *CreERT*, 75 mg/kg tamoxifen (Sigma, T5648) in corn oil (Sigma) was injected intraperitoneally into 8-week-old mice once a day, for 5 consecutive days. Rapamycin (R-5000, LC laboratories) was dissolved in ethanol and diluted in filter-sterilized vehicle (5% Tween-80, 5% PEG-40, 0.9% NaCl) immediately prior to intraperitoneal injection. Vehicle or 4 mg/kg rapamycin was administered every other day for 6 weeks. All animals were used under the NIH Guide for the Care and Use of Laboratory Animals and were handled according to the animal protocol approved by the University of Massachusetts Medical School Institutional Animal Care and Use Committee (IACUC).

## OVX-induced bone loss

To induce postmenopausal osteoporosis, 12-week-old female mice were anesthetized and bilaterally ovariectomized or subjected to a sham operation. All mice were then randomly assigned to treatment with vehicle or trametinib (0.6 mg/kg; *Hu-Lieskovan et al., 2015*). Inhibitor or vehicle treatment was given daily by oral gavage. After 8 weeks, mice were euthanized and subjected to bone analysis.

## Bone mechanical testing

To determine bone mechanical properties, 8-week-old male *Map2k2*<sup>–/–</sup> (Ctrl) and *Map2k1*<sup>fl/fl</sup>;*Dmp-1*<sup>Cre</sup>;*Map2k2*<sup>–/–</sup> (dKO<sup>Dmp1</sup>) femora were tested, as previously described (*Melville et al., 2014*). Right femora were loaded to failure in three-point bending in the anterior-posterior direction with a span length of 6 mm at a rate of 0.1 mm/s (858 Mini Bionix: MTS, Eden Prairie, MN, USA). Bending strength was calculated based on the force and displacement data from the tests, and midshaft geometry was measured with microCT.

## MicroCT

MicroCT analysis was performed as previously described (*Kim et al., 2020*). Sex-matched and control littermates were used, and analysis was performed by an investigator blinded to the genotypes of the animals under analysis. Femurs excised from the indicated mice were scanned using a microCT 35 (Scanco Medical) with a spatial resolution of 7 µm. For trabecular bone analysis of the distal femur, an upper 2.1 mm region beginning 280 µm proximal to the growth plate was contoured. For cortical bone analysis of femur and tibia, a midshaft region of 0.6 mm in length was used. MicroCT scans of skulls and lumbar vertebrae (L4) were performed using isotropic voxel sizes of 12 µm. 3D reconstruction images were obtained from contoured 2D images by methods based on distance transformation of the binarized images. All images presented are representative of the respective genotypes ($n > 5$).

## Histology, dynamic histomorphometry, and immunofluorescence

For histological analysis, hindlimbs were dissected from the mice, fixed in 10% neutral buffered formalin for 2 days, and decalcified by daily changes of 0.5 M tetrasodium Ethylenediaminetetraacetic acid (EDTA, pH 7.4) for 3–4 weeks. Tissues were dehydrated by passage through an ethanol series, cleared twice in xylene, embedded in paraffin, and sectioned at 7 µm thickness along the coronal plate from anterior to posterior. Decalcified femoral sections were stained with hematoxylin and eosin (H&E) or TRAP. To detect apoptotic osteocytes in bone tissue, terminal deoxynucleotidyl transferase

(TdT)-mediated dUTP-digoxigenin nick-end labeling (TUNEL) staining was performed using TUNEL assay kit (Abcam, ab206386) according to the manufacturer's instructions.

For dynamic histomorphometry analysis, 25 mg/kg calcein (Sigma, C0875) and 50 mg/kg alizarin-3-methyliminodiacetic acid (Sigma, A3882) dissolved in 2% sodium bicarbonate solution were subcutaneously injected into mice at 5-day intervals. After 2-day fixation in 10% neutral buffered formalin, undecalcified femur samples were embedded in methylmethacrylate (*Fukuda et al., 2013*). A region of interest was defined and bone formation rate/bone surface (BFR/BS), MAR, osteoblast surface/bone surface (Ob.S/BS), osteoclast number/bone parameter (N.Oc/B. Pm), osteoclast surface/bone surface (Oc.S/BS), erosion surface/bone surface (ES/BS), osteocyte number/bone area (N.Ot/B.Ar), and empty lacunae were quantitated using a semiautomatic analysis system (OsteoMetrics, Atlanta, GA, USA). Measurements were taken on two sections/sample (separated by ~25 μm) and summed prior to normalization to obtain a single measure/sample in accordance with ASBMR standards (*Dempster et al., 2013*; *Parfitt et al., 1987*). This methodology underwent extensive quality control and validation and the results were assessed by a research specialist in a blinded fashion.

For immunofluorescence, decalcified and cryo-sectioned samples were incubated with antibodies for CD31 (1:100, BD Pharmingen, 553370) and endomucin (1:100; Santa Cruz, sc-65495) were used as primary antibodies and Alexa Fluor 488 (1:400, Thermo, A21206) and Alexa Fluor 594 (1:400, Thermo, A11032) were used as secondary antibodies according to the manufacturer's instructions.

## CTx-I measurement

Serum level of CTx-I was measured according to the manufacturer's instructions (Immunodiagnostic Systems, AC-06F1).

## Transmission electron microscopy

TEM images were obtained from JEOL JEM 1400. For TEM analysis of femoral bone, whole femur was fixed with 4% paraformaldehyde in 0.08 M Sorenson's phosphate buffer at 4°C overnight and decalcified with 10% EDTA (pH 7.2) for 10 days. Decalcified bones were embedded in epoxy resin after dehydration through an ethanol series and transferred to a transitional solvent, propylene oxide. Ultrathin sections of metaphysis area were examined for osteocyte analysis (*Cheville and Stasko, 2014*).

## Transcriptome analysis

Eight RNA-seq samples (two GM control, two GM dKO, two OIM control, and two OIM dKO samples) were mapped to the mouse reference genome (Mus_musculus.GRCm38.80) with STAR aligner (v.2.6.1b; *Dobin et al., 2013*; *Dobin and Gingeras, 2015*). After mapping, read counts were generated by using HTSeq-count (v.0.11.3; *Anders et al., 2015*). The read counts were used for a differential expression analysis between control and dKO groups using DESeq2 (v.1.28.1; *Love et al., 2014*) with the ashr shrinkage estimator (v.2.2.47; *Stephens, 2017*). Statistically significantly expressed genes were determined as having absolute log-fold change larger than 1.5 and having a p value less than 0.005. For upregulated genes (log-fold change [LFC] >0; $n = 152$ for GM and $n = 2048$ for OIM) and downregulated genes (LFC <0; $n = 415$ for GM and $n = 1,576$ for OIM) in dKO samples, gene ontology (GO) analysis was performed with enrichGO function in clusterProfiler package (v.3.18.1; *Yu et al., 2012*) with the genome-wide annotation for mouse (org.Mm.eg.db; v.3.12) in Bioconductor. Category net plot is used for visualization using the cnet function in clusterProfiler. For gene set enrichment analysis (GSEA), differentially expressed genes in dKO RNA samples were analyzed using the GSEA software (v.4.1.0) with molecular signatures database (v.7.4) as annotation. Significant GO terms as adjusted p value <0.01 and q value <0.005 were identified for GO enrichment analysis. Nominal p value <0.01 and q value <0.25 were considered as a significantly enriched pathway with default settings for GSEA.

## Analysis of cell proliferation

Cell proliferation was assessed by alamarBlue staining (Thermo Scientific, DAL1100) or Bromodeoxyuridine (BrdU) incorporation assay (Abcam, ab126556).

**Table 1.** Primer sequences.

| Gene | Forward | Reverse |
|---|---|---|
| Human RUNX2 | TTACTTACACCCCGCCAGTC | CACTCTGGCTTTGGGAAGAG |
| Human SP7 | TTACAAGCACTAATGGGCTCCT | GTAGACACTGGGCAGACAGTCA |
| Human RPLP0 | GGAATGTGGGCTTTGTGTTC | TGCCCCTGGAGATTTTAGTG |
| Mouse Alpl | CACAATATCAAGGATATCGACGTGA | ACATCAGTTCTGTTCTTCGGGTACA |
| Mouse Ibsp | CAGGGAGGCAGTGACTCTTC | AGTGTGGAAAGTGTGGCGTT |
| Mouse Bglap | GCAGCACAGGTCCTAAATAG | GGGCAATAAGGTAGTGAACAG |
| Mouse Col1a1 | ACTGTCCCAACCCCCAAAG | ACGTATTCTTCCGGGCAGAA |
| Mouse Atf4 | CCACTCCAGAGCATTCCTTTAG | TGTCATTGTCAGAGGGAGTGTC |
| Mouse Map2k1 (MEK1) | ACTGCCCAGTGGAGTATTCAGT | TGTACCATGAGCTGCTTCAGAT |
| Mouse Map2k2 (MEK2) | CATCAGTGTAGGTCATGGGATG | AAACTCCTGGAAGTCTGAGCTG |
| Mouse Tnfsf11 (RANKL) | CAGCATCGCTCTGTTCCTGTA | CTGCGTTTTCATGGAGTCTCA |
| Mouse Tnfrsf11b (OPG) | CGGAAACAGAGAAGCCACGCAA | CTGTCCACCAAAACACTCAGCC |
| Mouse Dmp1 | GAAAGCTCTGAAGAGAGGACGG | CCTCTCCAGATTCACTGCTGTC |
| Mouse Phex | CTGGCTGTAAGGGAAGACTCCC | GCTCCTAAAAGCACAGCAGTGTC |
| Mouse Sost | CTTCAGGAATGATGCCACAGAGGT | ATCTTTGGCGTCATAGGGATGGTG |
| Mouse Gls | GGCAAAGGCATTCTATTGGA | CTTGGCTCCTTCCCAACATA |
| Mouse Pgc1a (Ppargc1a) | CCCACAACTCCTCCTCATAAAG | TTGGGTACCAGAACACTCACTG |
| Mouse Drp1 | GGGGTAAATTTCTTCACACCAA | TCAGGGCTTACCCCCTTATTAT |
| Mouse Mfn1 | ATAGAAGATGGCATGGGAAGAA | GCTGGAAGTAGTGGCTTCAAGT |
| Mouse Sgk1 | GGGTGCCAAGGATGACTTTA | TGGGTTAAATGGGGGTGTAA |
| Mouse Rplp0 | TGGCCAATAAGGTGCCAGCTGCTG | CTTGTCTCCAGTCTTTATCAGCTGCAC |
| Mouse Hk2 | GCCAGCCTCTCCTGATTTTAGTGT | GGGAACACAAAAGACCTCTTCTGG |
| mt-ND1 | CTAGCAGAAACAAACCGGGC | CCGGCTGCGTATTCTACGTT |
| mt-16srRNA | CCGCAAGGGAAAGATGAAAGAC | TCGTTTGGTTTCGGGGTTTC |

## Collection of CM and osteoblast and endothelial cell functional assay

CM was collected from control or dKO osteoblasts at day 6 of osteogenic culture. Briefly, cells were washed with PBS twice and incubated with serum free medium for 24 hr. Mouse EPOCs were obtained from BioChain (7030031) and cultured in GM (BioChain Z7030035). For capillary tube formation assay, EPOCs (30,000 cells/well) were seeded in a 96-well plate precoated with Matrigel (BD) and incubated with CM or FGF2. After 5 hr of incubation at 37°C, the number of tube branches in each well was quantified by counting four random fields per well as previously described (*Xu et al., 2018*). Alternatively, CM was added to osteogenic culture of wildtype BMSCs and osteogenic differentiation was assessed after 6 days of culture.

## Measurement of glutamate and ATP production

Extracellular glutamate release was determined according to the manufacturer's protocols (Promega, J8021). Alternatively, intracellular ATP levels were determined by reaction of ATP with recombinant firefly luciferase and its substrate D-luciferin according to the manufacturer's instructions (Thermo Scientific, A22066).

## Analysis of mDNA copy number

The mitochondrial copy number was determined by mtDNA/nDNA using real-time PCR for mt-ND1/Hk2 or mt-16srRNA/Hk2, as previously reported (*Quiros et al., 2017*). Sequences are provided in *Table 1*.

### Assessment of mitochondrial number and membrane potential

Cellular mitochondria were stained with Mito Tracker (Biotium, 70075) and detected by confocal microscopy (Leica SP8). Alternatively, flow cytometry was used to assess mitochondrial number and membrane potential of the cells treated with Mito Tracker (Biotium, 70075) and TMRE (tetramethyl-rhodamine ethyl ester, Biotium, 70005), respectively, according to the manufacturers' protocols. Flow cytometry analysis was performed on an LSR II (BD Biosciences).

## Measurement of OCR and intracellular ROS levels

Intracellular OCR (Cayman, 600800) and cellular reactive oxidative species (ROS) levels (C10422, Invitrogen) were measured according to the manufacturers' protocols.

### Luciferase reporter assay

Mouse calvarial osteoblasts were transiently transfected with RUNX2- (OG2-luc), β-catenin- (Topflash-luc), Smad- (3TP-luc), SP7- (SP7-luc), or ATF4-responsive (OSE1-luc) reporter gene along with *Renilla* luciferase vector (Promega) using the Effectene transfection reagent (Qiagen). After 48 hr, dual luciferase assay was performed according to the manufacturer's protocol (Promega) and luciferase activities were normalized to *Renilla*.

### RT-PCR and immunoblotting

To prepare bone RNA samples from mouse limbs, the hindlimbs were dissected and skin/muscle tissues were removed. The remaining tibias were chopped and homogenized. Total RNA from cells or tissues were extracted using QIAzol (QIAGEN) and cDNA was synthesized using the High-Capacity cDNA Reverse Transcription Kit from Applied Biosystems. Quantitative real-time PCR was performed using SYBR Green PCR Master Mix (Bio-Rad, Hercules, CA) with Bio-Rad CFX Connect Real-Time PCR detection system. mRNA levels were normalized to the housekeeping gene Ribosomal protein, large, P0 (*Rplp0*). The primers used for PCR are described in *Table 1*.

For immunoblotting, cell lysates were prepared in lysis buffer (50 mM Tris–HCl [pH 7.8], 150 mM NaCl, 1% Triton X-100, 1 mM dithiothreitol, 0.2% sarkosyl acid and protease inhibitor cocktail [Sigma, P8340]). Protein samples were subjected to sodium dodecyl sulfate polyacrylamide gel electrophoresis and transferred to Immobilon-P membranes (Millipore). Membranes were immunoblotted with the indicated antibodies and developed with ECL (Thermo Scientific). Immunoblotting with antibody specific to GAPDH was used as a loading control.

## Statistics and reproducibility

All experiments were performed a minimum of two to three times. For histological staining, flow cytometry, and immunoblotting, representative images are shown. All data are shown as the mean ± standard deviation (SD). We first performed the Shapiro–Wilk normality test for checking normal distributions of the groups. For comparisons between two groups, a two-tailed unpaired Student's *t*-test was used if normality tests passed and a Mann–Whitney test was used if normality tests failed. For the comparisons of three to six groups, we used one-way analysis of variance if normality tests passed, followed by Tukey's multiple comparison test for all pairs of groups. GraphPad PRISM software (ver.9.0.2, La Jolla, CA) was used for statistical analysis. $p < 0.05$ was considered statistically significant.

## Acknowledgements

We would like to thank Dr. Hwanhee Oh for help with the skeletal analysis. We also thank Oksun Lee and Jihea Kim for experimental support and the many individuals who provided valuable reagents. This project was supported by NIH-NIAMS R21AR077557 and AAVAA Therapeutics. M.B.G. holds a Career Award for Medical Scientists from the Burroughs Wellcome Fund, NIH support under award R01AR075585, a Novartis Institutes for Biomedical Research Global Scholars Award, and a Pershing Square Sohn Cancer Research Alliance award.

# Additional information

### Competing interests

Jae-hyuck Shim: is a scientific co-founder of the AAVAA Therapeutics and holds equity in this company. These pose no conflicts for this study. The other authors declare that no competing interests exist.

### Funding

| Funder | Grant reference number | Author |
|---|---|---|
| NIH-NIAMS | R21AR077557 | Jae-hyuck Shim |
| AAVAA Therapeutics | | Jae-hyuck Shim |
| Burroughs Wellcome Fund | | Matthew B Greenblatt |
| NIH-NIAMS | R01AR075585 | Matthew B Greenblatt |
| Novartis Institutes for Biomedical Research Global Scholars Award | | Matthew B Greenblatt |
| Pershing Square Sohn Cancer Research Alliance award | | Matthew B Greenblatt |

The funders had no role in study design, data collection, and interpretation, or the decision to submit the work for publication.

### Author contributions

Jung-Min Kim, Conceptualization, Data curation, Formal analysis, Validation, Investigation, Methodology, Writing – original draft, Writing – review and editing; Yeon-Suk Yang, Formal analysis, Validation, Investigation; Jaehyoung Hong, Hyonho Chun, Software, Formal analysis; Sachin Chaugule, Marjolein CH van der Meulen, Ren Xu, Formal analysis, Investigation; Matthew B Greenblatt, Conceptualization, Writing – review and editing; Jae-hyuck Shim, Conceptualization, Data curation, Supervision, Funding acquisition, Investigation, Writing – original draft, Writing – review and editing

### Author ORCIDs

Jung-Min Kim http://orcid.org/0000-0002-9072-4293
Marjolein CH van der Meulen http://orcid.org/0000-0001-6637-9808
Ren Xu http://orcid.org/0000-0001-6578-4553
Jae-hyuck Shim http://orcid.org/0000-0002-4947-3293

### Ethics

This study was performed in accordance with the recommendations in the Guide for the Care and Use of Laboratory Animals of the National Institutes of Health. All of the animals were handled according to approved Institutional Animal Care and Use Committee (IACUC) protocols (202200036) of the University of Massachusetts Chan Medical School. The protocol was approved by the Committee on the Ethics of Animal Experiments of the University of Massachusetts Chan Medical School (202200036).

### Decision letter and Author response

Decision letter https://doi.org/10.7554/eLife.78069.sa1
Author response https://doi.org/10.7554/eLife.78069.sa2

# Additional files

### Supplementary files

• Transparent reporting form

### Data availability

All data generated or analyzed during this study are included in the manuscript and supporting files. Source data files have been provided for all the figures.

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
