## [Editor Report]

This work provides a novel insight into regulation of osteogenesis by ERK-mTOR pathway. The authors proposed that the effect of Erk pathway would be mediated mTOR2-SGK1. The mitochondrial metabolisms appears to be involved in this regulation. This study is well performed, and the manuscript is clearly written.

---

## [Decision Letter]

**Decision letter after peer review:**

Thank you for submitting your article "Biphasic regulation of osteoblast development via the ERK MAPK-mTOR pathway" for consideration by *eLife*. Your article has been reviewed by 2 peer reviewers, and the evaluation has been overseen by a Reviewing Editor and Mone Zaidi as the Senior Editor. The reviewers have opted to remain anonymous.

Essential revisions:

1) As reviewer #1 mentioned, the authors should be more careful about the definition of "skeletal stem cells".

2) The mechanistic data or at least discussion would be required for the mechanisms underlying the increased osteoclast activity in the Dmp1-cre-dKO.

3) The expression level of each protein as well as its phosphorylated form needs to be shown in Figure 4 supplement 1, Figure 5 supplement 1, and Figure 6.

4) The authors should clarify how and/or which factors regulate bifunctional effects of MEK-Erk signaling in osteogenesis.

5) Discussions on the reviewers' questions (e.g. the physiological and pathological significance of osteoblast MEK-Erk axis, the mechanism underlying mTOR-PGK1 and/or mitochondrial signals regulate transcriptional events, and the different biological property of mTOR1 and mTOR2) should be provided.

*Reviewer #1 (Recommendations for the authors):*

I would raise a few concerns that need to be addressed prior to publication.

1. Although the authors focus on differentiation process from skeletal stem cells (SSCs) to bone-forming osteoblasts in the Introduction section, they did not specifically use the cell population in any assays in this work; they used hBMSCs for in vitro assays and DMP1-cre and Ocn-cre in vivo, both of which targeted late stages of the osteoblast lineage, not SSCs. Using "skeletal progenitors" may be more suitable rather than using SSCs in Introduction.

2. Page 10, lines 8 to 11: How does the increased osteoblast activity lead to the increased osteoclast activity in the Dmp1-cre-dKO? The authors are encouraged to provide mechanistic data or at least discuss the mechanism.

3. Figure 4 supplement 1, Figure 5 supplement 1, and Figure 6: To discuss the change of phosphorylation levels, expression level of each protein as well as its phosphorylated form needs to be shown.

4. Figure 6I: The authors are encouraged to show or discuss in more detail how ERK suppresses mTORC2 activity.

*Reviewer #2 (Recommendations for the authors):*

1. Although the finding of the role of MEK-Erk in the manuscript sounds important and new, the physiological and pathological significance were not shown and discussed. The reviewer believes that this is an important and interesting point for the Readers. Especially, human diseases such as osteoporosis could be discussed.

2. The reviewer is wondering how and/or which factors regulate bifunctional effects of MEK-Erk signaling in osteogenesis. The identification of the upstream keys strengthens the study.

3. Figure 3 A-D: The time course experiments are helpful.

4. RNA-Seq data are helpful for the Readers; however, how mTOR-PGK1 and/or mitochondrial signals regulate these transcriptional events. Osteogenic transcription factors, Runx2 and Sp7, are involved in these steps?

5. The reviewer notices that mTOR1 and mTOR2 have different biological property, although they share the same function in several pathways and biological function. The authors need to appreciate the point to describe them.

[Editors’ note: further revisions were suggested prior to acceptance, as described below.]

Thank you for resubmitting your work entitled "Biphasic regulation of osteoblast development via the ERK MAPK-mTOR pathway" for further consideration by *eLife*. Your revised article has been evaluated by Mone Zaidi (Senior Editor) and a Reviewing Editor.

The manuscript has been improved but there are some remaining issues that need to be addressed, as outlined below:

*Reviewer #1 (Recommendations for the authors):*

The authors have properly addressed the reviewer's concerns. The reviewer would support publication of this work.

*Reviewer #2 (Recommendations for the authors):*

The authors seemed to partly address the concerns raised by the reviewer. One of the major issues is clarifying the bifunctional effects of MEK/Erk signaling in osteogenesis. However, the reviewer was disappointed that the authors did not attempt to deeply chase the issues. The authors showed the potential involvement of Wnt and TGFβ signaling cascades in the steps based on RNA-seq analyses. The reviewer strongly suggests biochemically examining the effects of Wnt and TGFβ signaling in the cells.

---

## [Author Response]

Essential revisions:1) As reviewer #1 mentioned, the authors should be more careful about the definition of "skeletal stem cells".

As suggested by the reviewer #1, “skeletal stem cells” (SSCs) was changed to “skeletal progenitors” throughout the revised text.

2) The mechanistic data or at least discussion would be required for the mechanisms underlying the increased osteoclast activity in the Dmp1-cre-dKO.

As suggested by the reviewer #1, new data was added to the revised manuscript (revised Figure 2—figure supplement 1C), demonstrating reduced levels of OPG mRNA without any alteration in RANKL mRNA in the tibia of Dmp1-cre-dKO mice. These results suggest that an increase in the RANKL/OPG ratio is responsible for enhanced osteoclast activity in these mice.

3) The expression level of each protein as well as its phosphorylated form needs to be shown in Figure 4 supplement 1, Figure 5 supplement 1, and Figure 6.

As suggested by the reviewer #1, new immunoblot results showing the expression of each protein were added to the revised figures (Figure 3—figure supplement 4A, Figure 5—figure supplement 1, and Figure 6A).

4) The authors should clarify how and/or which factors regulate bifunctional effects of MEK-Erk signaling in osteogenesis.

To identify factors regulating the distinct roles of MEK/ERK MAPK signaling during osteogenesis, further transcriptional analyses, including gene-set enrichment analysis (GSEA), heatmaps of signature gene sets, and category net plots, were performed in *Ctrl* vs. *dKO* osteoblasts at both day 6 of osteogenic culture and under non-differentiating basal culture conditions (revised Figure 3—figure supplement 3). In this study, the genes associated with pro-osteogenic WNT/β-catenin and TGF-β signaling pathways were highly upregulated in mature *dKO* osteoblasts relative to *Ctrl* osteoblasts at the later day 6 differentiation stage. However, this pattern of gene expression was not detected under non-differentiating basal culture conditions reflective of an earlier stage of osteo-lineage differentiation. This suggests that, at late differentiation stages, impaired ERK activation upregulates WNT/β-catenin and TGF-β signaling leading to enhanced osteoblast differentiation. In contrast, these pro-osteogenic signaling pathways were not similarly upregulated by impaired ERK activation at early differentiation stages. Thus, WNT/β-catenin and TGF-β signaling is a candidate mediator of ERK’s divergent effects in the early vs late stages of osteoblast differentiation.

5) Discussions on the reviewers' questions (e.g. the physiological and pathological significance of osteoblast MEK-Erk axis, the mechanism underlying mTOR-PGK1 and/or mitochondrial signals regulate transcriptional events, and the different biological property of mTOR1 and mTOR2) should be provided.

As suggested by the editor, all of the reviewers’ questions were addressed in the Discussion section of the revised manuscript.

Reviewer #1 (Recommendations for the authors):I would raise a few concerns that need to be addressed prior to publication.1. Although the authors focus on differentiation process from skeletal stem cells (SSCs) to bone-forming osteoblasts in the Introduction section, they did not specifically use the cell population in any assays in this work; they used hBMSCs for in vitro assays and DMP1-cre and Ocn-cre in vivo, both of which targeted late stages of the osteoblast lineage, not SSCs. Using "skeletal progenitors" may be more suitable rather than using SSCs in Introduction.

As suggested by the reviewer, “skeletal stem cells” (SSCs) was changed to “skeletal progenitors” throughout the revised text.

2. Page 10, lines 8 to 11: How does the increased osteoblast activity lead to the increased osteoclast activity in the Dmp1-cre-dKO? The authors are encouraged to provide mechanistic data or at least discuss the mechanism.

We appreciate the reviewer for raising this point. Our new data demonstrated that OPG mRNA levels were markedly decreased in the tibial bone of the Dmp1-cre-dKO mice without any alteration in RANKL mRNA expression (revised Figure 2—figure supplement 1C), suggesting that upregulation of RANKL/OPG ratio may be responsible for enhanced osteoclast activity in these mice.

3. Figure 4 supplement 1, Figure 5 supplement 1, and Figure 6: To discuss the change of phosphorylation levels, expression level of each protein as well as its phosphorylated form needs to be shown.

As suggested by the reviewer, new immunoblot results showing the expression of each protein were added to the revised figures (Figure 4—figure supplement 1A, Figure 5—figure supplement 1, and Figure 6A).

4. Figure 6I: The authors are encouraged to show or discuss in more detail how ERK suppresses mTORC2 activity.

We thank the reviewer for raising this point. Previous studies have demonstrated that impaired ERK activation by the MEK inhibitor trametinib treatment resulted in elevated phosphorylation of the mTORC2 subunit Rictor or SIN1, activating the mTORC2 pathway in cancer cells (Cell Reports Medicine. 2020, PMID: 33294856; Science Signaling, 2017, PMID: 29184034). Further studies will be necessary to define how ERK can suppress mTORC2 activity in mature osteoblasts. A discussion of this point was added to the revised introduction and Discussion sections.

Reviewer #2 (Recommendations for the authors):1. Although the finding of the role of MEK-Erk in the manuscript sounds important and new, the physiological and pathological significance were not shown and discussed. The reviewer believes that this is an important and interesting point for the Readers. Especially, human diseases such as osteoporosis could be discussed.

As suggested by the reviewer, the overall contribution of the ERK MAPK pathway to physiologic development and pathologic disorders was described the revised introduction section.

2. The reviewer is wondering how and/or which factors regulate bifunctional effects of MEK-Erk signaling in osteogenesis. The identification of the upstream keys strengthens the study.

We thank the reviewer for raising this important point. -To identify factors regulating the distinct roles of MEK1/ERK MAPK signaling during osteogenesis, further transcriptional analyses, including gene-set enrichment analysis (GSEA), heatmaps of signature gene sets, and category net plots, were performed in *Ctrl* vs. *dKO* osteoblasts at both day 6 of osteogenic culture and under non-differentiating basal culture conditions (revised Figure 3—figure supplement 3). In this study, the genes associated with pro-osteogenic WNT/β-catenin and TGF-β signaling pathways were highly upregulated in mature *dKO* osteoblasts relative to *Ctrl* osteoblasts at the later day 6 differentiation stage. However, this pattern of gene expression was not detected under non-differentiating basal culture conditions reflective of an earlier stage of osteo-lineage differentiation. This suggests that, at late differentiation stages, impaired ERK activation upregulates WNT/β-catenin and TGF-β signaling leading to enhanced osteoblast differentiation. In contrast, these pro-osteogenic signaling pathways were not similarly upregulated by impaired ERK activation at early differentiation stages. Thus, WNT/β-catenin and TGF-β signaling is a candidate mediator of ERK’s divergent effects in the early vs late stages of osteoblast differentiation.

3. Figure 3 A-D: The time course experiments are helpful.

As suggested by the reviewer, new data showing the time course of ALP activity, cell viability, and osteogenic genes expression at day 0, 3, 6 of the osteogenic culture was added to the revised Figure 3—figure supplement 1D-F.

4. RNA-Seq data are helpful for the Readers; however, how mTOR-PGK1 and/or mitochondrial signals regulate these transcriptional events. Osteogenic transcription factors, Runx2 and Sp7, are involved in these steps?

We thank the reviewer for raising this interesting point. First, we examined effects of *Mek1/2*-deficiency on the expression and transcription activity of key osteogenic transcription factors, including RUNX2, SP7, or ATF4 at the late stages of osteoblast differentiation. Mature *dKO* osteoblasts displayed a significant increase in the expression and transcriptional activity of ATF4 (a master regulator of osteoblast maturation at the late stages), but not RUNX2 and SP7 (master regulators of osteoblast commitment at the early stages), suggesting that ERK’s roles in mature osteoblasts may be required for ATF4 regulation (Figure 3—figure supplement 5). Finally, we examined the effects of mTOR-SGK1 and/or mitochondrial signals on the expression and transcriptional activity of ATF4 in mature *dKO* osteoblasts. The enhanced expression and transcriptional activity of ATF4 in mature *dKO* osteoblasts were reversed by *Sgk1* knockdown or treatment with the mitochondrial inhibitor antimycin A, suggesting that mTOR-SGK1 and mitochondrial function control ERK-mediated regulation of ATF4 in mature osteoblasts (Figure 3—figure supplement 5D, E).

5. The reviewer notices that mTOR1 and mTOR2 have different biological property, although they share the same function in several pathways and biological function. The authors need to appreciate the point to describe them.

As suggested by the reviewer, a more detailed description of mTORC1 and mTORC2 in osteoblasts was added to the revised manuscript (page 17, line 3-12).

[Editors’ note: further revisions were suggested prior to acceptance, as described below.]

Reviewer #2 (Recommendations for the authors):The authors seemed to partly address the concerns raised by the reviewer. One of the major issues is clarifying the bifunctional effects of MEK/Erk signaling in osteogenesis. However, the reviewer was disappointed that the authors did not attempt to deeply chase the issues. The authors showed the potential involvement of Wnt and TGFβ signaling cascades in the steps based on RNA-seq analyses. The reviewer strongly suggests biochemically examining the effects of Wnt and TGFβ signaling in the cells.

As suggested by the reviewer, multiple biochemical analyses were performed to examine the effects of WNT and TGF-β signaling in Mek1/2 dKO osteoblasts. Consistent with transcriptome analysis showing upregulation of the genes associated with WNT/β-catenin and TGF-β/Smad signaling in dKO osteoblasts under osteogenic, but not undifferentiated, conditions (Figure 3- figure supplement 3A-C), luciferase assays using a β-catenin (Topflash-luc)- or Smad (3TPluc)- responsive reporter gene demonstrated a significant increase in transcription activities of WNT and TGF-β signaling in dKO cells under osteogenic, but not undifferentiated, conditions (Figure 3—figure supplement 4A). Likewise, protein levels of β-catenin and Smad3 were markedly increased in dKO osteoblasts cultured under osteogenic conditions (Figure 3-figure supplement 4B), suggesting that ERK inhibition at later stages of osteogenic differentiation enhances pro-osteogenic WNT and TGF-β signaling. Accordingly, pharmacological inhibition of WNT or TGF-β signaling reversed enhanced osteogenesis of dKO cells, as shown by reduced ALP activity and osteogenic gene expression in a dose-dependent manner (Figure 3-figure supplement 4C-F). These results suggest that enhanced WNT and TGF-β signaling at later stages of osteogenic differentiation of dKO osteoblasts might be major contributors to drive osteogenic differentiation program.